# Gold nanoparticles enhance antibody effect through direct cancer cell cytotoxicity by differential regulation of phagocytosis

Linyang Fan [1,2,4], Weizhi Wang[1,2,4], Zihua Wang[3] & Minzhi Zhao [1✉]

Ramucirumab is the first FDA-approved monotherapy for advanced gastric cancer. In this study, Ramucirumab (Ab) is attached to gold nanoparticles to enhance uptake efficiency. Gold nanoparticles can induce direct cytotoxic effects to cancer cells in the presence of Ab, while individual Ab or gold nanoparticles don't have such an effective anticancer effect even at extremely high concentrations. Proteomic and transcriptomic analyses reveal this direct cytotoxicity is derived predominantly from Ab-mediated phagocytosis. High affinity immunoglobulin gamma Fc receptor I shows differential up-regulation in gastric cancer cells treated by these nanodrugs compared with Ab, especially for Ab with gold nanorods. Simplified and powerful designs of smart nanoparticles are highly desired for clinical application. The enhancement of Ab accumulation with a simple composition, combined with direct cytotoxic effects specific to cancer cells brought improved therapeutic effects in vivo compared with Ab, which can promote further clinical application of gold nanomaterials in the diagnosis and therapeutics of gastric cancer.

[1] CAS Key Laboratory for Biomedical Effects of Nanomaterials and Nanosafety, CAS Center for Excellence in Nanoscience, National Center for Nanoscience and Technology of China, Beijing 100190, China. [2] School of Chemistry and Chemical Engineering, Beijing Institute of Technology, Beijing 100081, China. [3] Fujian Provincial Key Laboratory of Brain Aging and Neurodegenerative Diseases, School of Basic Medical Sciences, Fujian Medical University, Fuzhou 350122 Fujian, China. [4]These authors contributed equally: Linyang Fan and Weizhi Wang. ✉email: xgt1986627@163.com

The Gastric Cancer (GC) is one of the most severe malignant cancer with a high morbidity and lethality in China[1] and all over the world[2] GC has been top 5 morbidity in USA for more than 40 years. 5-year relative survival rate was merely 30%[3]. In China, the incidence and mortality reached the 2nd among all kind of tumor. Unfortunately, most diagnosed GC patients had been already in advanced and inoperable status[4] and chemotherapy would be the last hope for them. However, it is an undeniable fact that 5-year relative survival rate was actually double by modern diagnosis and tumor therapies development in 40 year since several gastric relative targets had been discovered, like Epidermal growth factor receptor (EGFR)[5], Human epidermal growth factor receptor 2(HER2)[6], Vascular endothelial growth factor(VEGF) family[7], Mesenchymal-epithelial transition factor(MET)(8) and Programmed death1(PD1): PDL (PD-ligand) 1/PDL2 pathway[8]. But among these targeted drugs described above, Cetuximab (anti-EGFR)[9], Rilotumumab (anti-cMET)[10] were all failure in clinical research, while OPDIVO (anti-PD-1) was failed in GC and Keytruda (anti-PD-1) was failed in 2rd line trail by KEYNOTE-061 trail[11]. Nowadays, only trastuzumab(anti-HER2), ramucirumab (anti-VEGFR2) and PD-1(although OPDIVO and Keytruda were finally approved, their clinical trails results were not all satisfactory enough as described) were the merely approved drugs[12,13]. 7–30% GC could be benefit from trastuzumab[14,15,34] while 36–40% GC could treated by ramucirumab[16,17]. And Ramucirumab had advantages in trastuzumab resistant patients[18]. Ramucirumab targeted to VEGFR2, a member of VEGF family and had a better prognosis in REGARD[19] and RAINBOW[20] trial. According to this trial, patients who received ramucirumab plus chemotherapy had a median overall survival of 9.63 months compared with 7.36 months for those in the control[12]. Hence, antibody-drug conjugate (ADC) has been considered as a better therapy method[21]. Ramucirumab was the only target drug approved by FDA for advanced or metastatic gastric or gastroesophageal junction adenocarcinoma as monotherapy[22].

Nanomaterial-based drug delivery has been widely applied in biomedical research field including photothermal therapy, imaging and drug delivery[6,23]. It was feasible to achieve high functional ligand densities on the surface for targeting purposes because of the high surface area to volume ratio of nanoparticles[10]. Utilize functionalized nanoparticles with biomolecules that target unique or overexpressed biomarkers in tumor cells to enhance delivery efficiency. Gold nanoparticle, as the well-known nontoxic biocompatible metal, was an especially promising candidate for cancer theranostic[24]. Due to tunable localized surface plasmon resonance (LSPR), gold nanorod are not only attractive probes for cancer cell imaging but they can also become highly localized heat sources when irradiated with a laser through the photothermal effect and can be used to provide hyperthermal cancer therapy, as well as to trigger drug release for chemotherapeutics. Thus, gold nanorod (AuNR) was selected for major investigation in the nano-delivery platform[25,26], and gold nanosphere (AuSP) was set as control.

In this work, to establish a more efficient drug delivery platform, gold nanorod (AuNR) was chosen, linking by polyethylene glycol (PEG) to antibody Ramucirumab (Ab) and a chemotherapy drug Doxorubicin (DOX), which was a chemotherapy drug and applied in GC for more than 20 years[27,28]. The preparation process of AuNR-PEG-Ab-DOX was showed in Fig. 1. The physical and biological effects of these nano-platforms were evaluated in vitro and in vivo respectively.

## Results and discussion
### Synthesis and characterization of gold nanomaterial delivered drugs. The composite materials were characterized (Fig. 2). There

were obvious capsules on the surface of gold nanorods (AuNR) and nanosphere (AuSP) characterization by TEM (200,000×), which confirmed the successful coating on the surface of the gold nano-particles (Fig. 2a, b). The capsule of nanoparticles was thicker when antibody was added into materials. Figure 2c, d showed that the synthesis of all kind of materials was accomplished by spectrophotometer detection. And spectrum exhibited red shift when Au modified by PEG. When Au-PEG linked to Ab, the sharp of spectrum changed.

Absorbances of the C-O unit at 1000–1200 cm$^{-1}$, the carbanyl unit at 1700 cm$^{-1}$ and hydroxy group at 3300–3400 cm$^{-1}$ in the Fourier-transform infrared (FTIR) spectrum demonstrated the conjugation of Hetero-functional PEG (α-Mercapto-ω-carboxy PEG solution, HS-C2H4-CONH-PEG-O-C3H6-COOH, MW. 3.5 kDa) (Supplementary Fig. 3). Conjugation Ab was also verified by observing the increase in hydrodynamic diameters (given as mean values from the number distribution) with dynamic light scattering (DLS) because IgG molecule increases the overall size of a nanoparticle conjugate significantly, as well as the changes of their surfaces charge (Supplementary Table 1).

### In vitro and in vivo imaging for the uptake and accumulation efficiency of nano-drugs. To evaluate the tumor cells recognition, combination speed and the uptake efficiency of nanoparticles, a low concentration of antibody (1 µg/mL in Ab, FDA recommend 8 mg/kg, equal to 8 µg/mL) was applied, and confocal laser scanning microscopy (CLSM; 63×, oil-immersion objective) imaging and flow cytometry (FCM) were employed (Fig. 3a–e). GC cell SNU-5 were treated by Ab, PEG-Ab, AuSP-PEG-Ab, or AuNR-PEG-Ab (1 µg/mL Ab) for 1, 2 and 4 h. The fluorescence in cells was increasing from Ab to AuNR-PEG-Ab group and cells treated by Ab for 4 hours were still appeared vague. The similar phenomenon was exhibited in AuSP-PEG-Ab and AuNR-PEG-Ab treated cells, as the triple fluorescence intensity than Ab by FCM measurement at 1 h. In fluorescence imaging results, 2 hour's treatment of AuNR and AuSP performed equally to PEG-Ab treated for 4 h and much better than Ab. What's more, AuNR and AuSP contained nanodrugs were obviously endocytosed by cells in 4 h, as Ab and PEG-Ab were still stained encircle the cells. To test the gathering effect of Ab in vivo, all drugs were intravenous injected into tumorigenic BALB/c-nu mice (Fig. 3f–h). AuNR-PEG-Ab was injected at 30 µg Ab, meanwhile Ab at 30 µg and 5× at 150 µg according to human recommended dose 8 mg/kg[22,29]. No tumor mouse injected by Ab showed that the majority of drugs were metabolic located in liver. To evaluate drug recognition ability, fluorescence was measured in tumor and normal tissues by Image J software (1.48 v). The biodistribution of the excised organ fluorescent images showed that Ab (30 µg) performed poor and there was no significant change of fluorescent intensity until the quality of Ab increasing to 5-fold. AuNR-PEG-Ab (30 µg) performed better than Ab (30 µg) and 5×Ab (150 µg) group. AuNR-PEG-Ab group provided similar performance in all groups at 2 hours detection, while there was a significant increase at 8 hours and max fluorescent intensity at 24 h in AuNR-PEG-Ab group (Fig. 3g). What's more, AuNR-PEG-Ab's $F_{ratio}$ raised 10 times than non-tumor and Ab (30 µg) in 24 h (Fig. 3h).

### Analysis of toxic effects of different nano-carriers and nano-drugs at cellular level. VEGF/VEGFR2 (KDR) is a classical interaction signal pathway in HUVECs[30]. VEGF could induced HUVECs to overexpress VEGFR2 and promote angiogenic sprouting, endothelial cell differentiation and permeability directly[31,32]. Besides, the VEGF/VEGFR-2 system played an important role in survival, proliferation and anti-apoptosis in many kinds of cancer cells[33]. We used 20 ng/mL VEGF to

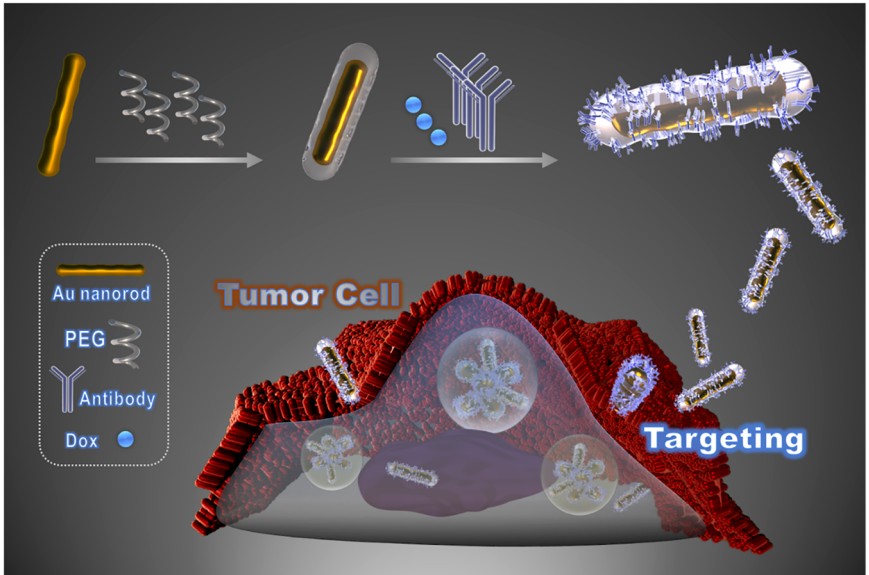

**Fig. 1 Scheme.** The fabrication and drug delivery of AuNR-PEG-Ab-DOX.

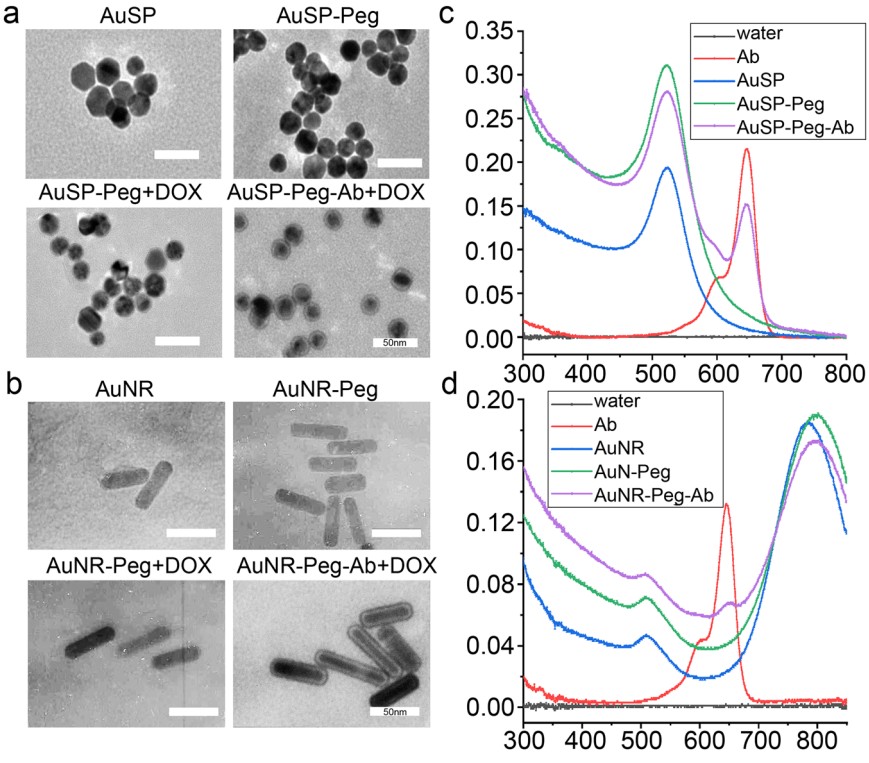

**Fig. 2 Gold nanoparticles characterization by TEM and spectrum. a**, **b** Representative images of TEM detection for gold nanosphere and nanorods (AuSP and AuNR), Au-PEG, Au-PEG+DOX with or without linking by Ramucirumab (Ab). **c**, **d** Spectrum measured red shift when Au modified by PEG and Ab.

stimulate HUVECs as a VEGFR2 reliable cell model and VEGFR2 cell affinity validation assay showed cell toxicity equaled to 6.6 µg/mL (C value in 4 parameter curve) and fulfilled FDA requirement as 8 mg/kg (Fig. 4a). However, Ab cannot provide equal effect for SNU5, MKN-45 (another gastric cancer cell) and GES-1 (normal stomach cell) in our research (Fig. 4b–d). In detail, in SNU5, MKN-45 and GES-1, cell viability had no change in $10^{-2}$–$10^{3}$µg/mL of Ab, whereas HUEVC cell results formed a normal S sharp. In a word, the Ab treatment alone hadn't taken significant influence to cell viability of stomach tumor cell and normal stomach cell.

Our nanoparticles contained AuNR/AuSP, PEG, antibody and DOX. AuNR/AuSP was the core of the complex. PEG played a role of linker between Au and Ab/DOX. Ab played a role of scout for searching target cells and DOX played a role of soldier for enhancing toxicity. To enhancing cytotoxin efficiency of Ramucirumab. The involvement of gold nanorod and DOX beside Ab could act as the combining of target therapy, chemotherapy and phototherapy, which would synergistically abrogate tumors and prevent their recurrent, either with or without tumor resection[25]. To be specific, cell viability tests were carried out to the series of nano-vehicles for evaluating the cytotoxicity to GC cells. In this

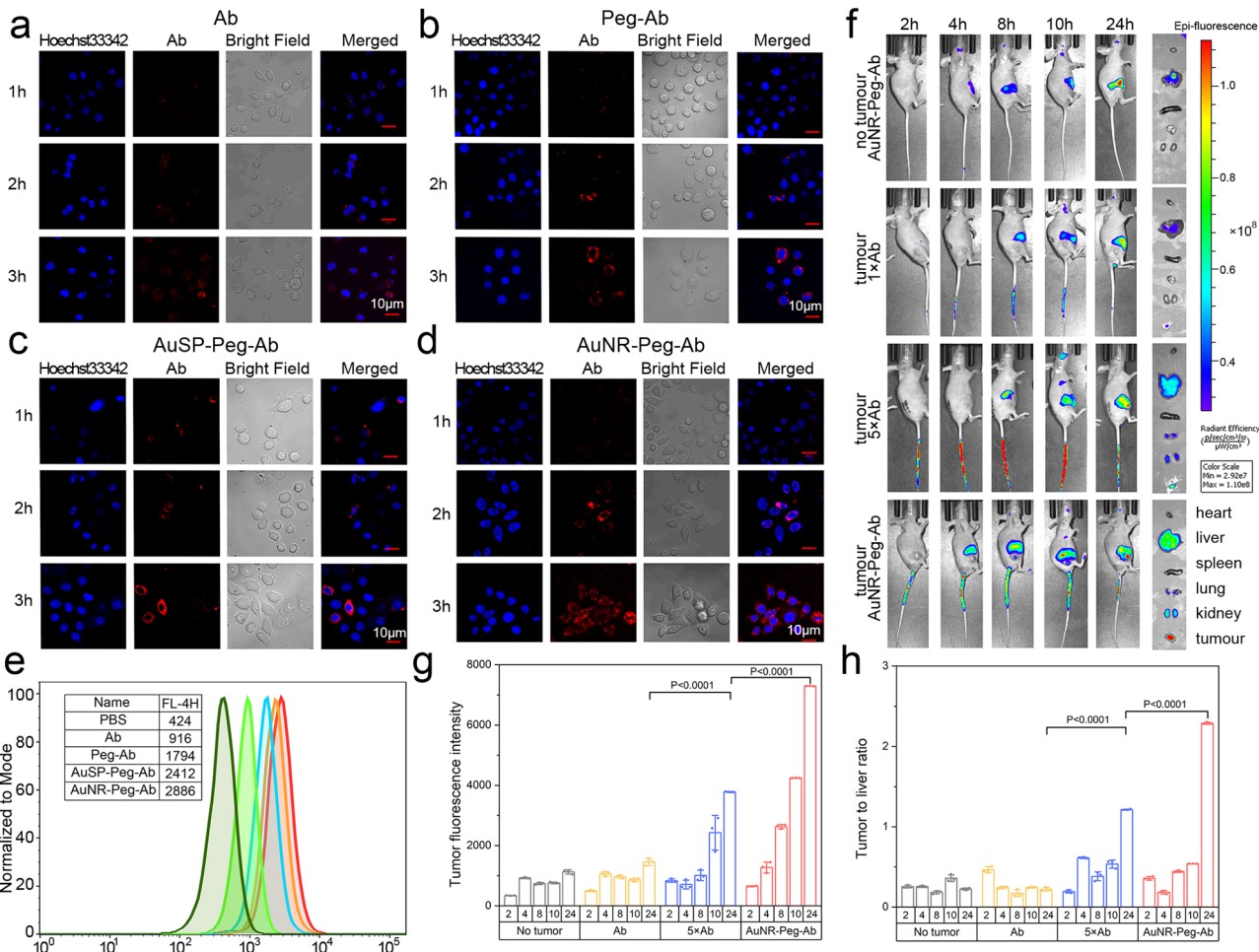

**Fig. 3 Tumor-targeting capability of Ab and Au-PEG-Ab. a–d** Representative immunofluorescence images of **a** Ramucirumab (Ab), **b** Ab linked to PEG, **c**, **d** Gold nanoparticles (AuSP and AuNR) linked Ab treated SNU5 cells. **e** Flow cytometry analysis of nanoparticles cellular uptake efficiency. **f–h** In vivo imaging of the gold nanomaterial in the SNU5 tumor bearing mice. **f** Images of tissue and major organs at different time periods after intravenous injection of Ab and AuNR-PEG-Ab. **g**, **h** fluorescent intensity ($F_{tumor}$) and $F_{ratio}$ ($F_{tumor}$ to $F_{Liver}$ ratio) measurement at different time. Data represent mean ± SD ($n = 3$). Statistical difference was evaluated by one-way ANOVA followed by Tukey's post hoc test.

study, Ab take the core effect in target therapy. So the final loading concentrations of Ab contained in nano-carriers was chosen also as 8 mg/kg (8 µg/ml) per FDA requirement equally in each group. And the final concentration of other reagents like PEG, gold nanoparticles and DOX could be calculate by the coupling ratio of them with Ab (see the detail value of each reagent in the method section). Then the difference of cytotoxicity for each group would not be attributed to the amount of Ab within the complex. The results indicated that neither gold nanomaterials nor other linkers had cell inhibition effect, except for DOX (Fig. 4e, i, k). When these components were formed as Au-PEG-DOX, it inhibited cell viability prominently. It was interesting that the nano-carriers can strengthen the inhibition of cell viability when linked to Ab, no matter the existing of DOX or not. Furthermore, the promotion effect was significantly stronger in the AuNR-PEG-Ab treated cells than AuSP-PEG-Ab group (Fig. 4j, l, $p < 0.05$). This distinction hadn't seen in single gold or Au-DOX group. In order to demonstrate whether this difference was only showed in SNU5 cell, MKN45 cell (another GC cell) was tested and the results were the same. In apoptosis analysis (8 µg/mL Ab), both AuSP-PEG-Ab and AuNR-PEG-Ab could induce SNU-5 cells annexin V(apoptosis maker) exstrophy. Furthermore, Q3 cell ratio by AuNR-Ab was more than AuSP-Ab, hinting that the

different shape of material caused diverse physiological effects (Fig. 4m). It was the first time that we found AuNR based nano-carriers had shape-dependent inhibition of gastric cancer cell viability in the presence of Ab. The recent report for toxicity was not for sphere or rod shape[34]. The earlier studies on shape-dependent toxicity of gold nanomaterials which were focused on spherical- and rod-shaped nanoparticles did not discussed the shape[35–38]. The only one for shape effect of toxicity which indicated that spherical gold nanoparticles were generally more toxic than rod-shaped particles suggested that the higher toxicity of CTAB-coated spheres came from a higher release of toxic CTAB upon intracellular aggregation[39]. Our results were different from all investigations discussed above. Our gold nanoparticles contained no CTAB per manufacture's instruction, and neither rod- nor spherical-shaped nanoparticles didn't show any toxicity effect to SNU-5 cells when they were treated alone. The coating used in this study was PEG, which was regarded as biocompatible and its toxicity was also confirmed in our results (Fig. 4e, i, k). And PEG-Ab also showed nearly no toxicity (Fig. 4i, k). In addition, the AuNR and AuSP themselves, as well as combined with DOX, showed no difference on toxicity in two GC cells. Thus, this shape-depend cell toxicity had close relation with Ab.

Our results had proved that the intracellular concentration of Ab in GC cell was nearly the same for both AuNR-PEG-Ab (NR

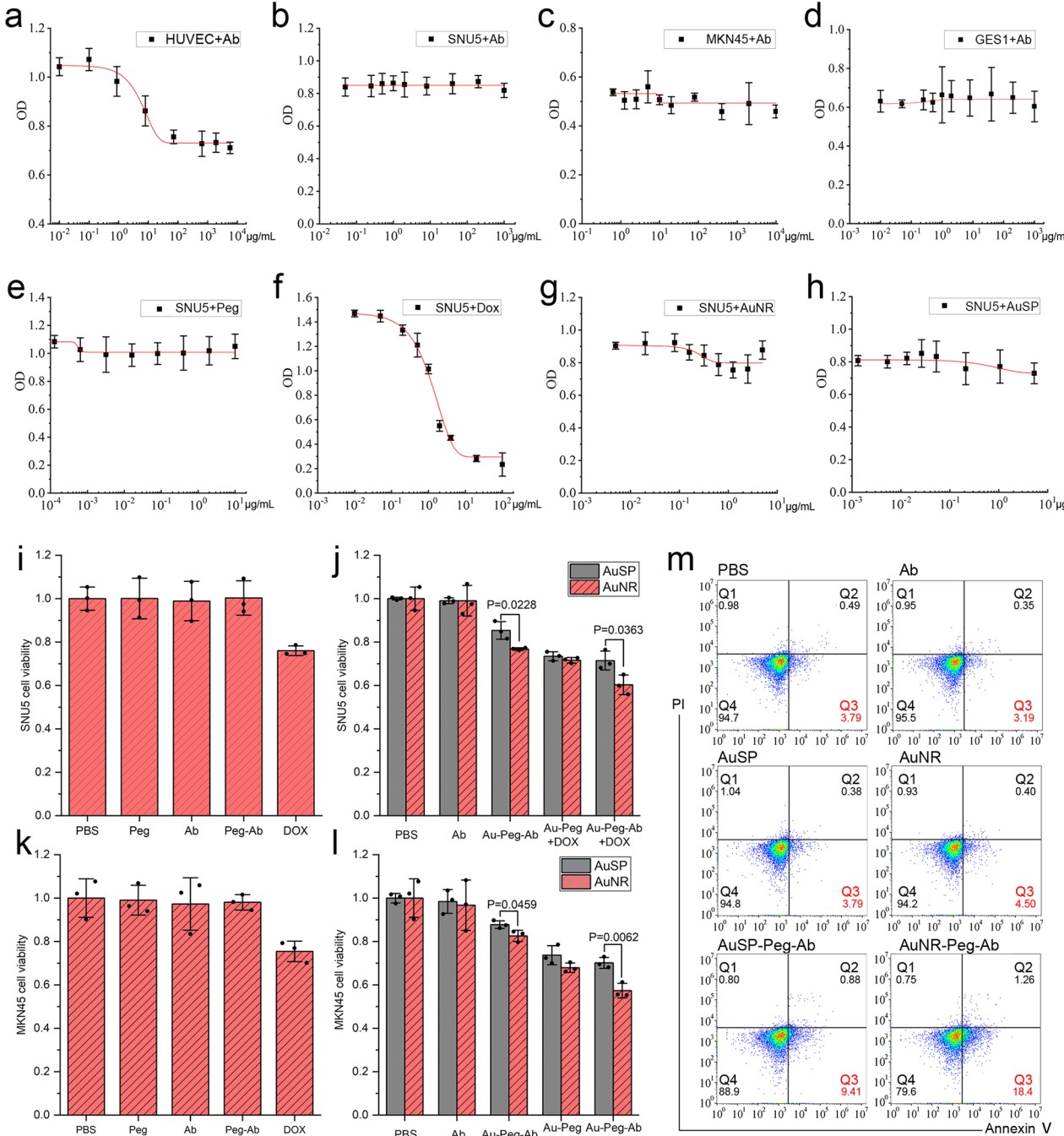

**Fig. 4 Cytotoxicity tests in different cells. a–d** IC$_{50}$ curve treatment by Ramucirumab (Ab) in HUVEC cells (**a**), SNU5 cells (**b**), MKN-45 cells (**c**), and GES-1cells (**d**). **e–h** IC$_{50}$ curve treatment by PEG (**e**), doxorubicin (DOX) (**f**), AuNR (**g**) and AuSP (**h**) in SNU5 cells. **i–j** MTT measurements of cell viability in SNU5 cells treated with different components. **k**, **l** MTT measurements of cell viability in MKN-45 cells treated with different components. **m** Cell apoptosis analysis by flow cytometry. Data represent mean ± SD ($n = 3$). Statistical difference was evaluated by unpaired, two-tailed independent Student $t$-test.

group) and AuSP-PEG-Ab (SP group) (Fig. 3c–e), as well as Ab itself didn't influence the cell viability/toxicity (Fig. 4a–d). Although the existing of gold nanomaterials enhanced the accumulation and uptake of Ab, but the most aggregation enhancement came from PEG (Fig. 3b, e), which hadn't induced inhibition of cell viability as PEG-Ab (Fig. 3i, k). Then it could be deduced that the reason for the difference might not mainly come physically from the mount of Ab, but from the biological interaction of gold nanomaterials with Ab.

**Quantitative proteomics and transcriptomics analysis for the mechanism of the cytotoxicity diversity between NR group (AuNR-PEG-Ab) and SP group (AuSP-PEG-Ab).** In order to discover the biological mechanism for the difference of GC cell viability at molecular level, mass spectrometry based quantitative proteomics and RNA-sequencing based transcriptomics strategies were employed on NR group (AuNR-PEG-Ab) were compared with SP group (AuSP-PEG-Ab). The results were shown in Fig. 5 and Supplementary Dataset 1-2.

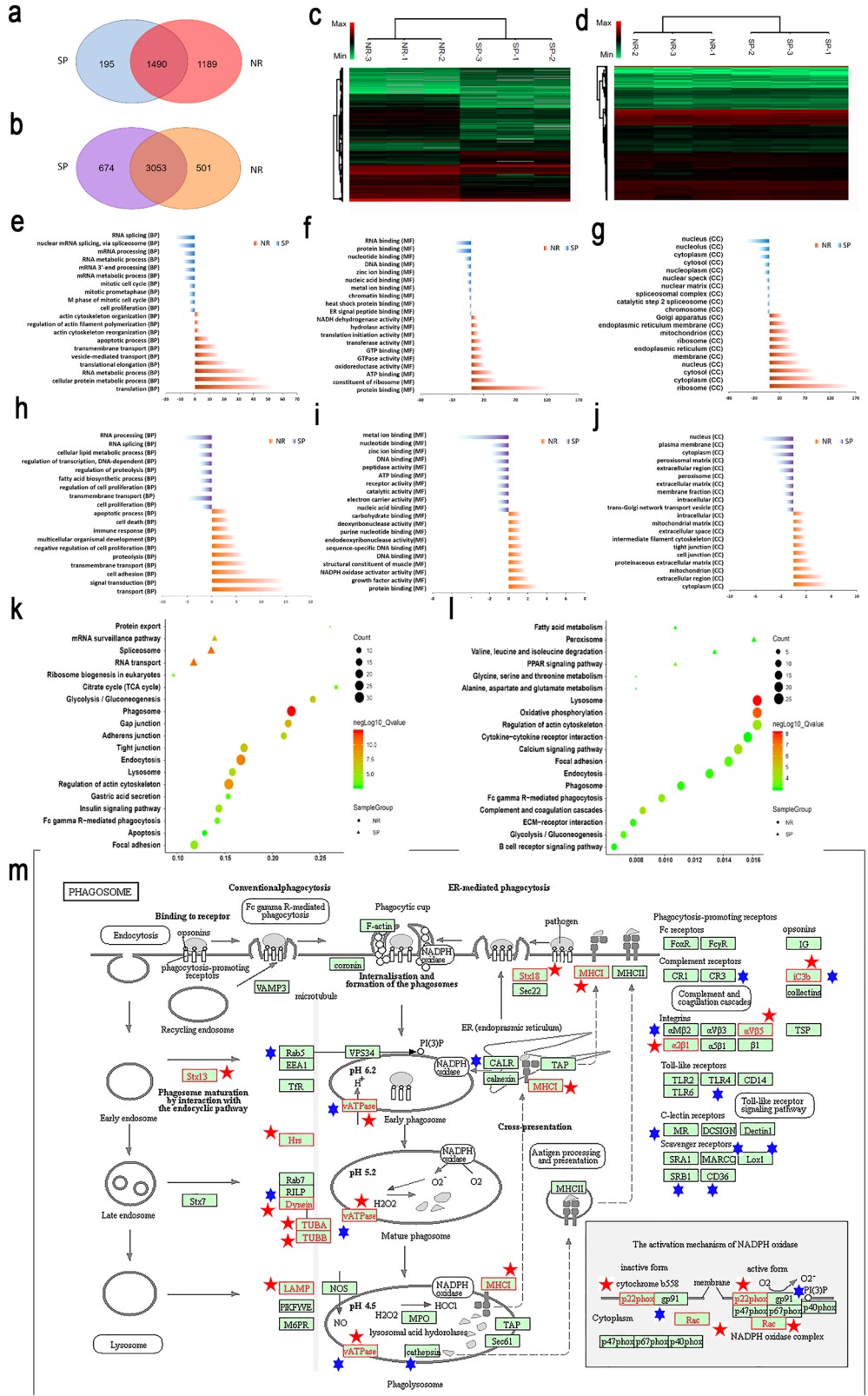

The number of altered proteins and expressed genes only in NR group were 1189 and 501, while 195 and 674 in SP group, with 1490 proteins and 3053 genes overlap (Fig. 5a, b). Although with similar number of altered genes, the number of altered proteins was obviously more in NR group. Hierarchical clustering for these differential expression protein and gene sets was created with an absolute log2 fold-change cutoff of ≥1 and an adjusted p value (q value) of ≤0.05. Large amount of proteins and differential expressed genes were derived from the different nano-drugs. It can be seen that the biological replications were clustered together and different groups were separated well. In addition, the pattern of the mRNA level was relative uniformity for NR and SP group, while in the protein level the difference was more apparently. Furthermore, the functional enrichment was analyzed via Gene

**Fig. 5 Quantitative proteomics and transcriptomics analysis of AuNR-PEG-Ab (NR group) compared with AuSP-PEG-Ab (SP group) in SNU5 cells. a, b** Venn grams of the whole numbers of differential proteins and expressed genes quantified in proteomics and transcriptomics from different groups. **c, d** Differential protein and expressed gene heat maps for NR and SP groups. **e–g** Top 10 items in the Gene Ontology (GO) biological process (BP), molecular function (MF), and cellular component (CC) enrichment of differential proteins in the proteomic results. **h–j** Top 10 items in the GO BP, MF, CC enrichment of differential expressed genes in the transcriptomic results. **k–l** KEGG enrichment of differential proteins and expressed genes. the color indicated the level of q value and the items with q < 0.001 were included. **m** Pathway "phagosomes" from KEGG with altered proteins (red pentagram) and expressed genes (blue hexagram) mapped in it. Statistical difference of differential expression protein and gene sets were created by unpaired, two-tailed independent Student t-test with an adjusted p value (q value) of ≤0.05. (n = 3).

Ontology (GO) biological process (BP), molecular function (MF), cellular component (CC) and Kyoto Encyclopedia of Genes and Genomes (KEGG) pathway. The proteins and genes which presented only in and up-regulated in NR group were recognized as NR altered, and the ones which presented only in and up-regulated in SP group were recognized as SP altered. Figure 5e–g showed the top 10 enriched items in GO o proteomic results and Fig. 5h–j showed the top 10 enriched items in GO from transcriptomic results. There were several items displayed similar in both protein and mRNA level. For example, apoptotic process (BP), transport (BP), and protein binding (MF) appeared in NR group. RNA splicing (BP), cell proliferation (BP) metal ion binding (MF) and nucleus (CC) appeared in SP group. Together with cell death (BP) in NR group and cell cycle related process of BP in SP group, it showed that NR group possessed more inhibition effect to cell viability relative to SP group. It can be seen that translation (BP) showed predominantly enrichment in proteomic results, which suggested that there were many differences coming from translation, the step after transcription. This can also explain the reason that there were comparable differential expression genes for both NR and SP group in transcriptomic results, but more proteins distinct for NR group. The apoptotic process (BP) enriched in NR group, which displays one source of the more inhibitory effect of AuNR to tumor cells.

Then the enrichment for KEGG signal pathways were analysis from proteomic and transcriptomic level (Fig. 5k–l). There were more pathways enriched in NR group than SP group for both levels. "Phagosome" and "Lysosome" were the most enriched categories at proteomic and transcriptomic level. Phagocytosis plays an essential role in host-defense mechanisms through the uptake and destruction of pathogens[40]. Cross-linking of Fc of Ab initiates a variety of signals mediated by tyrosine phosphorylation of multiple proteins membrane remodeling to the formation of phagosomes. Nascent phagosomes undergo a process of maturation that involves fusion with lysosomes.[41,42] "Phagosome" showed obvious enrichment at both proteomic and transcriptomic level, and the members enriched in this pathway were labeled in Fig. 5m. Some other immune related items like "Fc gamma R(FcγR)-mediated phagocytosis" which related with antibody directly, as well as "Complement and coagulation cascades" and "B cell receptor signaling pathway" were also enriched in NR group. Multiple mechanisms have been observed for cancer therapy by monoclonal antibody, including the Fc-dependent effector mechanisms, complement dependent cytotoxicity (CDC), and Ab-dependent cellular cytotoxicity (ADCC)[43]. The FcγR, B cell and complement related pathways were enriched, which indicated the relation of difference had certain correlation with Ab function and immune response, which also showed in GO BP result of NR group at transcript level.

**Gold nanocarrier induced the direct cancer cell cytotoxicity by the immune related phagocytose mediated by differential regulation of FcγR CD64 on cancer cell.** There were only tumor cells existed in this study. It was reported that antibody may induce programmed cell death (PCD) directly correlated with reactive oxygen species (ROS). This pathway evokes PCD directly in the target cell in an actin-dependent, lysosome-mediated process. ROS generation was mediated by nicotinamide adenine dinucleotide phosphate (NADPH) oxidase[44]. Antibody mediated PCD pathway can be enhanced via Fc cross-linking by secondary Antibody or FcγR-expressed cells[43]. These features were fitted to the significant enriched pathways in our study, like "Phagosome", "Regulation of actin cytoskeleton", "Lysosome", "Oxidative phosphorylation" and "Endocytosis". Some categories in the GO BP and GO MF enrichments also had relation with this, like "NADPH oxidase activator activity"[45]. ROS generation was also detected (Fig. 6a). The results showed that the ROS value in Au-PEG-Ab was higher than individual Ab or Nano-materials. Furthermore, NR group presented higher value than SP group, which could be one of the reasons for a higher cell death rate of NR group.

Except for professional phagocytes including macrophages, neutrophils, and monocytes, epithelial cells, fibroblasts and other cells can take up particles[46]. Phagosomes can also be produced in non-professional phagocytes. Since fibroblasts lack Fc receptors, the antibodies could not act as an opsonins but only take effect through a common 'non-immunological' mechanism. The cell type used in the mechanism study was tumor cell, not fibroblasts or epithelial cells. There isn't any report said that tumor cell was phagocyte. Even if tumor cell could also be regarded as non-professional phagocytic cell, tumor cell hasn't been realized to possess immune related function or has immunoreceptor. However, it could be seen obviously about the immune process and immunoreceptor related enrichments in our proteomic and transcriptomics results, like "Phagosome" and "Fc gamma R(FcγR)-mediated phagocytosis". Thus, the expression levels of Fc gamma receptors, High affinity immunoglobulin gamma Fc receptor I (CD64) and Low affinity immunoglobulin gamma Fc region receptor III-A (CD16)[47], were detected from the mRNA and protein level (Fig. 6d, e; left). The results showed that CD64, high affinity receptor, had noticeable expression in tumor cells, and CD16 didn't show any expression. The Au-PEG-Ab groups which containing both Ab and nanomaterials showed significant up-regulation than the left groups. It was up-regulated in NR group compared with SP group (p < 0.05). And the Ab treated group showed similar expression level with the groups only treated with nanomaterials. This was in accordance with the cell viability results of tumor cells (Fig. 4i–l). Gastric cancer cell endogenously express Fc gamma receptor (FcγR) may facilitate the cytotoxicity, which might be the important reason that gave Au-PEG-Ab groups higher cell death effect of tumor cell even without the help of immune system. The expression situation of CD64 in normal gastric epithelial cells (GES-1) were also been tested. The results displayed that none of these group showed any expression (Fig. 6d, e; right). We supposed that the strong endogenous expression of cancer cell was an important reason for the cytotoxicity of these Ab containing nanodrugs. Thus, we further investigated the cell viability of GES-1 cell. The results in Fig. 6b, c verified that all

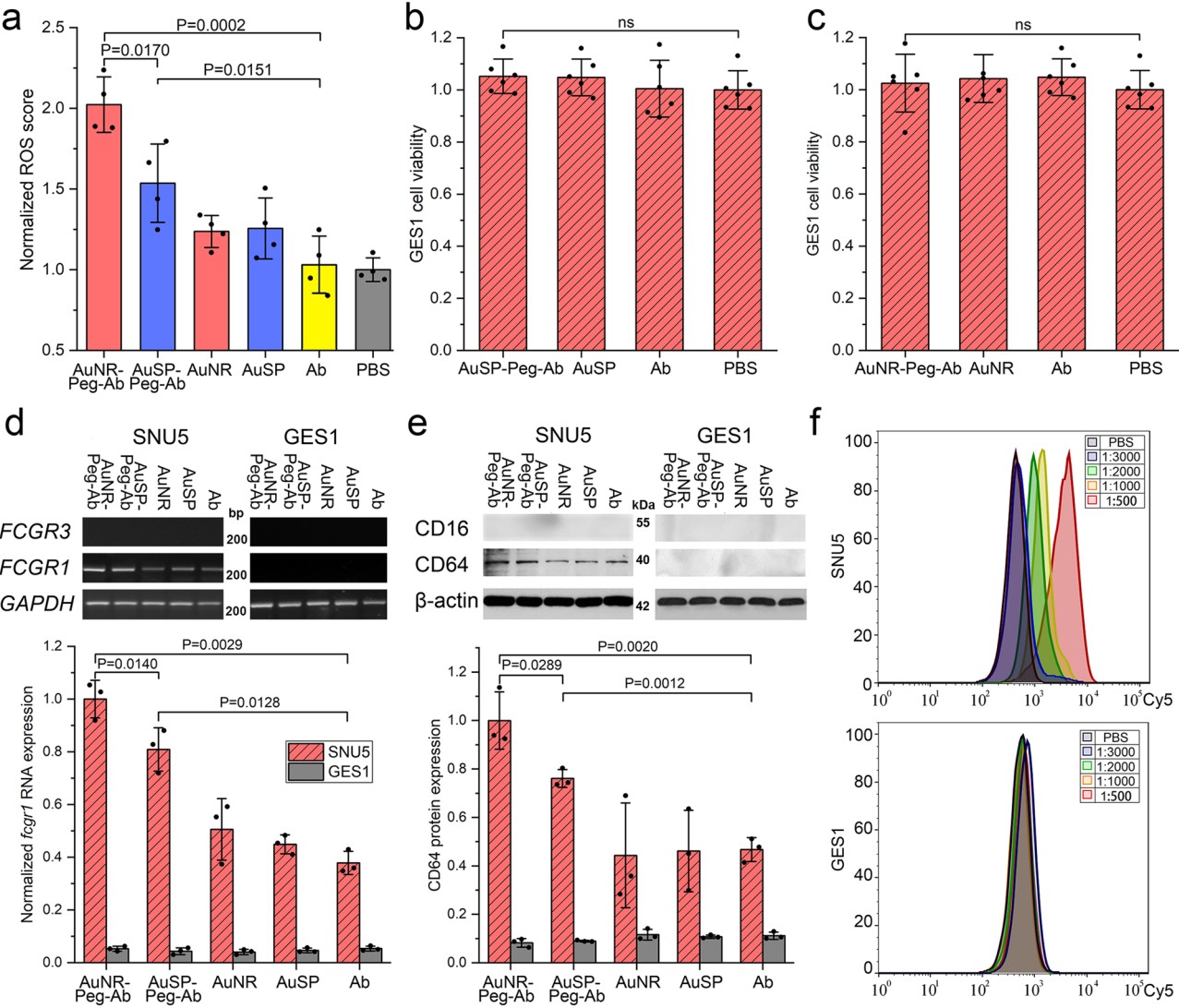

**Fig. 6 The potential course of gold nanoparticles induced tumor cytotoxicity. a** ROS generation test of SNU5 cell. Data represent mean ± SD ($n = 4$). **b**, **c** Cell viability analysis in GES-1 cell treated by different material. Data represent mean ± SD ($n = 6$). **d**, **e** mRNA and protein expression level CD64 in SNU-5 and GES-1 cells. Data represent mean ± SD ($n = 3$). **f** Flow cytometry analysis of Ab in SNU5 and GES1 cells. PEG was omitted in the mark of the nanodrug complex. Statistical difference was evaluated by one-way ANOVA followed by Tukey's post hoc test.

these treatments didn't possessed any cell cytotoxic effect to GES-1cell compared with control. It was interesting that the gold nanomaterials could activated the endogenous expression of FcγR specifically in gastric cancer cell when it was linked with Ab, meanwhile Ab couldn't achieve this alone. From the cell toxicity results Fig. 4j, l it can be seen that SP group also induced tumor cell inhibition, and SP group also induced some regulation of CD64 compared with the left groups except for NR group. To explore whether SP group can also evoke such immune related response, the transcriptome of SP group (AuSP-Ab) and NR group (AuNR-Ab) were analyzed respectively, both were compared with no treat group. The results showed in Supplementary Fig. 5 indicated that both of them aroused up-regulations correlated with the immune related processes and the antibody directly related pathway like "Fc gamma R(FcγR)-mediated phagocytosis" and "Phagosome". However, NR group triggered more enrichments than SP group, and NADPH related process only showed up-regulation in NR group. This is the reason that NR group showed up-regulation in these pathways and processes in the direct comparison to SP group.

According to the results of flow cytometry analysis to the different concentration of Ab treated SNU5 and GES-1 cells (Fig. 6f), SNU5 cells could be a VEGFR2 high expressed cells. And the uptake of Ab increased with the rising of the added Ab amount. Meanwhile, the cell death effect was not influenced by Ab (Fig. 4b). GES-1 cell might not uptake any Ab even at a high added concentration, or it doesn't expression VEGFR2. This might be the reason that the nano-carrier groups (NR and SP) couldn't induce cell death of GES-1 cells (Fig. 6b, c), and it was another superiority for target therapy to gastric cancer because this direct inhibition doesn't take effect in normal stomach cell.

This meant that AuNR could induce more cell death than AuSP specific to stomach cancer cell via the activation of immune receptor and immune phagocytose related pathway without the assistance of immunologic cell. The most important was that Ab must be contained but couldn't achieve this itself even at an extremely high concentration. These results suggested the superiority of AuNR and it would be benefit for the clinical application of gold nanorod in the immunologic therapy, especially for gastric cancer (as another tumor cell like Hela doesn't express VEGFR2

or doesn't uptake this Ab, which would not evoke the direct tumor cell killing effect, see Supplementary Fig. 6).

**In vivo analysis of nano-drugs in inhibiting tumor growth.** On the base of in vitro tests, the immunologic therapy effect in vivo was carried out on SNU5 xenografted mouse model. 20 mice were randomly and equally divided into four groups ($n = 5$). When the tumors had been developed to approximately 100–200 mm³, mice were injected intravenously once a week (day 1 and day 7) with AuNR-PEG-Ab (NR group), AuSP-PEG-Ab (SP group), Ab and PBS (100 µL) at a dose corresponding to 8 mg/kg of Ramucirumab. The tumor sizes and mice weights were recorded daily at the same time (Fig. 7a). After that, tumors and other organs were collected. The toxicity in tumors and organs of each group were determined by H&E. The apoptosis of tumor cells was also detected by TUNEL assay. As shown in Fig. 7a, the tumor growth rate of the NR and SP groups were more inhibited obviously compared with the other two groups. And the Ab treated group only showed slightly higher inhibitory effect than PBS treated group. NR group displayed the best therapy effect in all of the four groups with comparable body weight (Fig. 7a, b). Tumor and main organs were excised after the treatment. The hematoxylin-eosin (H&E) stained tumor from the NR group showed significant damages in tumor tissue. Besides, all the immunologic therapy groups hadn't showed obvious damage to normal organs for the morphologies of heart, liver, spleen, lung, and kidney had no obvious difference to that of the PBS treated group (Fig. 7c). The results were further confirmed by the terminal deoxynucleotidyl transferased dUTP Nick-End labeling (TUNEL) assay (Fig. 7d). NR and SP groups showed obvious cell apoptosis degree, and NR possessed more.

These founding in this study had two meanings. First, the gold nanorod can induced stronger biological response not because carrying more antibody, but via evoking response of target gastric cancer cell and inducing molecular function directly. What's more, this promotion effect can take effect only with the existing and assistance of antibody. Thus, this unique function of gold nanorod is specific to antibody-targeted therapy, which means that neither chemotherapeutic drug nor small molecular targeted drugs can get this effect. No matter for the advance of nano-delivery system, or for the advance of antibody-targeted drug themselves, these results would give more incites for further exploring in the biological effect of nanomaterial and antibody interactions.

In summary, we focused on promoting the effect of the promising targeted monoclonal antibody drug Ramucirumab (Ab) for GC, by the gold nanorod with perfect physicochemical and biocompatible properties. It was found that antibody could enhance the recognition, uptake and accumulation efficiency by nano delivery system in vitro and in vivo. Furthermore, the nanodrug could induce cell death effect in GC cells directly but not in normal gastric cell, compared with Ab and nanomaterial themselves. According to proteomic and transcriptomic results, the cytotoxic effects mainly came from Fc gamma receptor-mediated phagocytosis. Furthermore, it was found that *Fcgr1*(CD64), the high affinity receptor gene for Fc gamma, was up-regulated in nanodrugs, especially for AuNR-PEG-Ab group. Its high uptake and gathering, as well as direct cytotoxicity specific to GC cells explained its excellent therapeutic effect in vivo. The simple delivery system was more beneficial for the application of nanomaterial in clinic for better safety and controllability, and these findings will give more references for the better application of gold nanorod in the drug-delivery therapy in gastric cancer.

## Methods

**Design of gold particles-PEG conjugate to Cy5-labeling of Ramucirumab.** Ramucirumab (Cyramza, Lilly), HS-PEG3500-COOH (Nanosoft), Doxorubicin (Zhejiang Hisun Pharmaceutical Co) were purchased from manufactures. Heterofunctional PEG (α-Mercapto-ω-carboxy PEG solution, HS-C₂H₄-CONH-PEG-O-C₃H₆-COOH, MW. 3.5 kDa) link to 9–15 nm diameter, 46–56 nm length gold nanorods (Alfa Aesar 46819, AuNR) in 18 MEG DI water mixed with SDS for 12 h in RT. PEG excess was removed by centrifugation when PEG was quantified by the Ellman's assay to calculate the concentration whether PEG was all/half link to gold nanoparticles. 200 µL of PEG solution in 100 µL of phosphate buffer (0.5 M, pH 7) with 7 µL 5,5'-dithio-bis(2-nitrobenzoic) acid (DTNB, 5 mg/mL) in phosphate buffer (0.5 M, pH 7) and measuring the absorbance at 412 nm after 10 minutes reaction.10 kDa dialysis to remove PEG as to preparation amount of Au-PEG. Optimized ratio with gold nanorods to PEG is 1 mL:0.14 mg as all of PEG link to gold nanorods. Lower ratio of Ramucirumab to Alexa Fluor® 647(Ex: 650 nm, Em: 670 nm) to ensure there were Cy5-free primary amines (R-NH₂) of antibody. Ramucirumab functionalized with Au-PEG by standard EDC/NHS coupling reaction[48]. Gold nanospheres (STREM 95-1547, AuSP) replaced nanorods to figure out whether material shape affect cytotoxin performance. Briefly, 300 µL Au-PEG was activated by 2 mg/mL N-hydroxysulfosuccinimide (sulfo-NHS, Sigma) and 0.5 mg/mL EDC (1-Ethyl-3-(3-dimethylaminopropyl) carbodiimide, Sigma) with 400 µL 10 mM MES (2-(N-morpholino) ethanesulfonic acid, Sigma) for 1 h and purified by dialysis. Then, 100 µL activated Au-PEG incubated in 30 µL Ab-Cy5(1 mg/mL, totally 0.2nmol) or 10 µL DOX (0.1 mg/mL, totally 2nmol), and 100 µL Au-PEG activated with 15 µL Ab-Cy5(1 mg/mL, totally 0.1nmol) and mixed 5 µL DOX (0.1 mg/mL, totally 1nmol) to synthesis Au-PEG-Ab-DOX, and DOX was mixed directly in the system. 220–850 nm Spectrum and TEM validate that Au-PEG, Au-PEG-Ab, Au-PEG-DOX, Au-PEG-Ab-DOX were synthesis correctly. 20 nm Gold nanospheres (STREM 95–1547) replaced nanorods to figure out whether material sharp affect cytotoxin performance. All groups were quantified by the same Ab concentration (1 µg/mL in Ab).

**Ellman's assay.** The excess of thiolate chains in the supernatant was quantified by interpolating a calibration curve set by reacting 200 µL of PEG solution in 100 µL of phosphate buffer (0.5 M, pH 7) with 7 µL 5,5'-dithio-bis(2-nitrobenzoic) acid (DTNB, 5 mg/mL) in phosphate buffer (0.5 M, pH 7) and measuring the absorbance at 412 nm after 10 minutes reaction.

**Materials characterization.** 220–850 nm UV-vis Spectrum, TEM, Dynamic light scattering (DLS) and Infrared spectra (IR) validate that Au-PEG, Au-PEG-Ab were synthesized correctly. TEM (Tecnai G-20 spirit BioTwin, FEI, USA) was used to characterize the size, shape and morphology of the nanocarriers. After functionalization, the samples were collected and washed with PBS for three times. In the following, they were fixed with a 2% glutaraldehyde at 4 °C overnight, followed by postfixing for 2 h with 1% osmium tetroxide. The samples were dehydrated with graded ethanol and then infiltrated and embedded in Spurr's resin. Thin sections were mounted on copper grids and then observed on TEM. DLS measurements were performed with a Malvern Zetasizer. DLS measurements were carried out in water and hydrodynamic diameters, given as mean values from the number distribution. Surface charges were measured while the nanoparticles were dispersed in deionized water. The errors are standard errors ($n = 3$). FTIR analyses were carried out using a NICOLET iN 10 MX system. Reflectance micro-Fourier Transmitted Infrared spectra were collected over the 4000 to 1500 cm⁻¹ wavenumber range, at a resolution of 4 cm⁻¹.

**Cell culture and cell viability analysis.** HUVEC cells (Endothelial Cell Medium, supplemented with 5% fetal bovine serum), GES-1(1640 Cell Medium, Gibco, supplemented with 10% FBS), SNU5(1640 Medium with 10% FBS) and MKN-45 cell (1640 Medium with 20% FBS) was purchased from ATCC. Firstly, we established a VEGF-reliable cellular growth model in HUEVC cell, made VEGF protein as a stimulate factor to HUVEC cell activity. 9000 cells seed in ever well in 96-well plate with Endothelial Cell Medium. VEGF added into medium as a final concentration of 20 ng/mL in the next day. After 96 hours of incubation, cell was starvation by basal Endothelial Cell Medium overnight, Ramucirumab diluted by Endothelial Cell Medium supplemented with 5% fetal bovine serum to final concentration on a range of $10^{-2}$–$10^4$µg/mL and mix into per well for 24 hours. The concurrent method was applied into SNU5. While doxorubicin treatment to SNU5, tumor cells were seed into 96 well plate by 5000 per well, basal 1640 serum change preceding 10% FBS-1640 serum after 24 h, Ramucirumab diluted by 5% FBS-1640 serum at a final concentration of $10^{-3}$-$10^3$ µg/mL and change preceding culture serum. MTT assay detection after treatment for 24 h. MKN45 cell is another poorly differentiated gastric carcinoma cells. 3-(4,5-dimethyl-2-thiazolyl)-2,5-diphenyl-2-H-tetrazolium bromide (MTT) assays evaluate each drugs' toxicity in inhibition ratio. Data were collected by PerkinElmer Enspire Multiscan Spectrum. Statistics and significance analysis by OriginPro 2018 and SPSS 19 software. Then we compared the toxicity among each components for 24 h, such as PBS(NC), PEG(1 µg/mL), Ab (8 µg/mL), PEG-Ab (8 µg/mL in Ab), AuSP-PEG-Ab (8 µg/mL in Ab), AuSP-DOX (0.3 µg/mL in DOX), AuSP-Ab-DOX(8 µg/mL in Ab), AuNR-PEG-Ab(8 µg/mL in

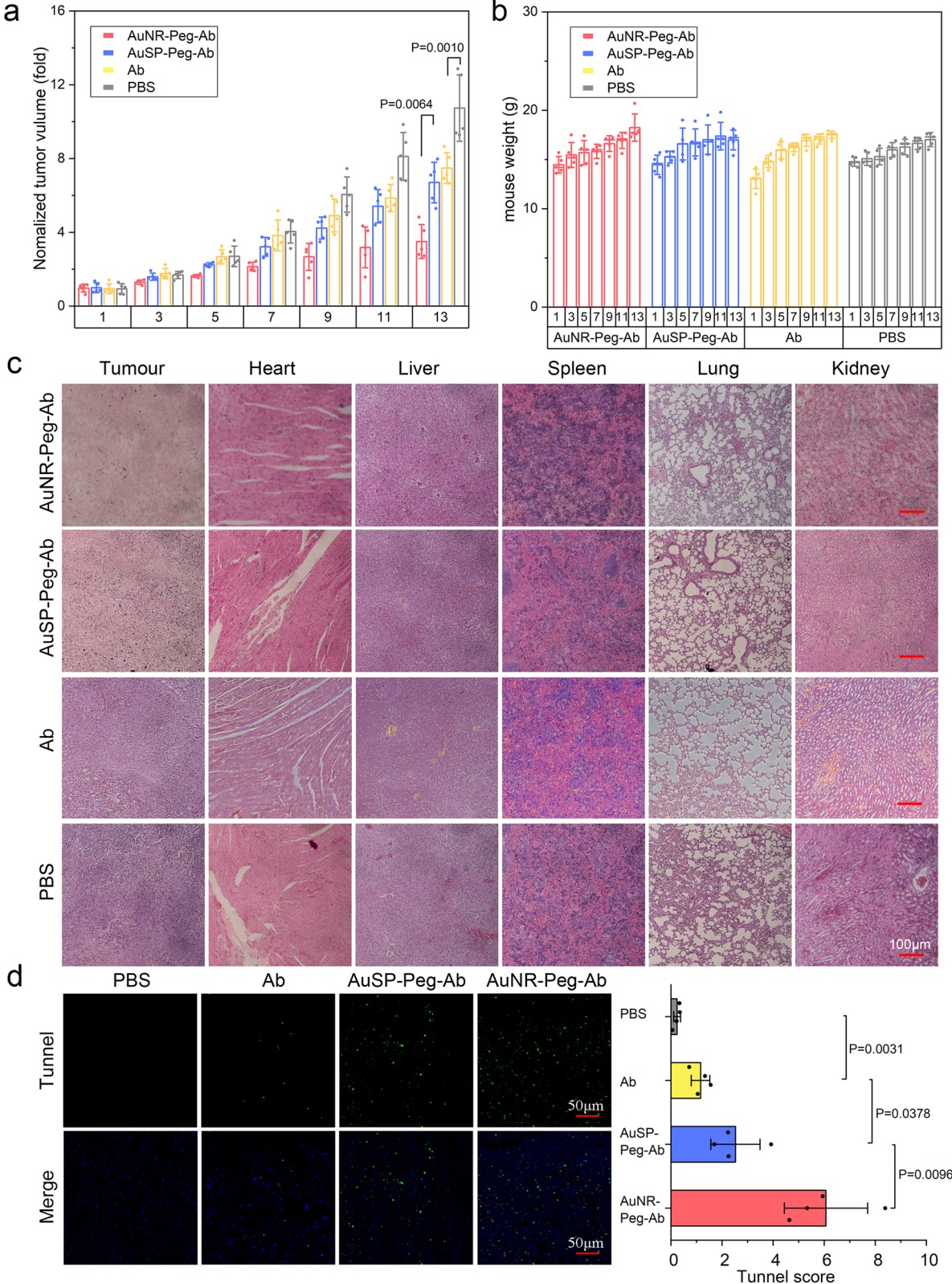

**Fig. 7 In vivo evaluation Ab connected nanomaterials in tumor inhibition. a** The changes in tumor size during the curing process. Data represent mean ± SD ($n = 5$). **b** The changes in mice body weight during the curing process. Data represent mean ± SD ($n = 5$). **c** Representative images of H&E images of the tumor and other tissue slices stained with H&E. **d** Tunnel immunofluorescence straining detected apoptosis cells. Data represent mean ± SD ($n = 4$). Statistical difference was evaluated by one-way ANOVA followed by Tukey's post hoc test.

Ab and 0.3 µg/mL in DOX), AuNR-DOX(0.3 µg/mL in DOX), AuNR-Ab-DOX(8 µg/mL in Ab and 0.3 µg/mL in DOX).

**In vitro fluorescence imaging**. SNU5 cells were cultured in the medium at the density of $10^5$ per confocal dish. Ramucirumab was labeled by Alexa Fluor® 647 (Ex: 650 nm, Em: 670 nm) and subsequently synthesis AuNR-PEG-Ab-Cy5 And AuSP-PEG-Ab-Cy5. All compounds treated SNU5 cells for 1, 2, 4 hours at final antibody concentration of 1 µg/mL. Hoechst 33342 (2 µg/mL, 200 µL, Ex: 405 nm, Em: 488 nm) was used as the nuclei indicator for 15 min incubation at 37 °C. Finally, all cells imaging were obtained by Zeiss LSM 710 confocal microscope.

**Animals and xenograft tumors**. Athymic female BALB/cAnNCrl nude mice (6-weeks old) were purchased from Charles River (Wilmington, MA, USA) and maintained on a 16:8 h of light-dark cycle under temperatures of ~18–23 °C with 40–60% humidity. All procedures were in accordance with the Beijing University Animal Study Committee's requirements and approved by the institutional Animal Study Committee of the National Center for Nanoscience and Technology. To obtain xenografts in the nude mice, SNU5 cells were trypsinization and suspended in 1× PBS and injected subcutaneously at the right dorsal side of the upper hindlimb of mice ($1 \times 10^7$ cells per mouse).

**In vivo fluorescence imaging**. When the tumor volume reached ~100 mm³ in long diameter, AuNR-PEG-Ab (30 µg Ab), Ab (30 µg) and 5×Ab (150 µg) were degermed by 0.22µm filter membrane and intravenous injected in each mouse. Fluorescence imaging was performed at 2, 4, 8, 12, 24 h after injection, using a PerkinElmer (Waltham, MA, USA) IVIS instrument with excitation at 650 nm and an 680 nm filter to collect the fluorescence emission of Cy5. Mice were euthanized, the major organs (liver, lung, spleen, kidney and heart) and tumor were dissected and imaging were obtained.

**Flow cytometry analysis**. The cellular uptake of Cy5-labeled Ab and other nanodrugs were investigated by a flow cytometry analysis. Cells were incubated with Ab for 1 min at room temperature. Then these cells were washed with PBS and re-suspended in PBS for analysis. The Cy5 fluorescence intensity (4 channel) was measured with a flow cytometer (accuri C6, BD, USA). Cy5-labeled Ab (1 mg/mL) was diluted by PBS (1:500, 1:1000, 1:2000, 1:3000). HUVEC and SNU5 cells incubated in Ab for 1 h at RT. While SNU5 cells incubated in Ab, PEG-Ab, AuSP-PEG-Ab and AuNR-PEG-Ab (1 µg/mL Ab) for 1 h at RT. Then these cells were washed with PBS and re-suspended in PBS for analysis. The Cy5 fluorescence intensity (4 channel) was measured with a flow cytometer (Accuri C6, BD, USA) using CFlowPlus software. Gating cells by FSC and SSC and $10^4$ cells data were collected and export to.fcs data. Then FLowJo V10 software was used to analyze.fcs data and re-gating cells by FSC and SSC to sort approximately $10^4$ cells, analyze cell's Cy5 fluorescence intensity by channel 4.

For cell apoptosis assays, SNU5 cells were plated overnight. Nanodrugs (NC refers to PBS) diluted by culture medium at the same concentration of MTT assay and they were treated cells for 24 h. Then cells were collected by trypsinization without EDTA, and $10^6$ cells were doubly stained with Annexin-V- Alexa Fluor 647/PI for 15 min and analyzed. After sorted approximately $10^4$ cells by FSC/SSC. Channel 3 and channel 4 was exhibited in a picture. PI and Alexa Fluor647 Quad gating according to NC group result. then we used the same condition to Quad gate all group.

**Protein digestion and LC–MS/MS analysis**. SNU5 cell samples were added with lysis buffer containing 50 mM Tris-HCl (pH 8), 8 M Urea and 0.2% SDS. Then they incubated with ultrasonication on ice for 5 min and then centrifuged at 13000 g for 20 min at 4 °C. The supernatant was transferred to a clean tube, and protein concentration was determined with a BCA assay. Add 2 mM DTT and incubate the sample at 56 °C for 1.5 hour following with adding sufficient iodoacetic acid and incubating for 1.5 hour protected from light at room temperature. And then add 4 volumes cold acetone to a sample extract, vortexed well, placed sample at −20 °C overnight. Centrifuge and collect pellet to wash twice with cold acetone. Finally dissolve the pellet by dissolution buffer containing 0.1 M triethylammonium bicarbonate (pH 8.5) and 8 M urea. Then, samples were digested with Trypsin Gold (Promega) at 37 °C for 20 hours. After trypsin digestion, peptide was desalted with C18 cartridge then dried by vacuum centrifugation. Peptides were fractionated using a C18 column (Waters BEH C18 4.6×250 mm, 5 µm) on a Rigol L3000 HPLC operating at 1 mL/min. The column oven was set as 50 °C. Mobile phases A (2% acetonitrile, adjusted pH to 10.0 using ammonium hydroxide) and B (98% acetonitrile, adjusted pH to 10.0 using ammonium hydroxide) were used to develop a gradient elution. The solvent gradient was set as follows: 3%B, 5 min; 3–10% B, 0.1 min; 10–20% B, 11.9 min;20–32% B, 11 min; 32–45% B, 7 min; 45–80% B, 3 min; 80% B, 5 min; 80–5%, 0.1 min; 5% B, 6.9 min. The tryptic peptides were monitored at UV 214 nm. Eluent was collected every minute and then merged to 15 fractions. The samples were dried under vacuum and reconstituted in 0.1% (v/v) formic acid (FA) in water for subsequent analyses.

An EASY-nLCTM 1200 UHPLC system (ThermoFisher) coupled to an Orbitrap Q Exactive HF-X mass spectrometer (ThermoFisher) were used for proteomics. Peptides were reconstituted in 0.1% FA was injected onto an Acclaim

PepMap100 C18 Nano-Trap column (2 cm×100 µm, 5 µm). Peptides were separated on a Reprosil-Pur 120 C18-AQ analytical column (15 cm×150 µm, 1.9 µm), using a 90 min linear gradient from 5 to 100% eluent B (0.1% FA in 80% ACN) in eluent A (0.1% FA in H₂O) at a flow rate of 600 nL/min. The solvent gradient was set as follows: 5–8% B, 2 min; 8–35% B, 80 min; 35–56% B, 2 min; 56–89% B, 1 min; 89–100% B, 5 min. Mass spectrometer was operated in positive polarity mode with spray voltage of 2.3 kV and capillary temperature of 320 °C. Full MS scans from 400 to 1500 m/z were acquired at a resolution of 60000 (at 200 m/z) with an AGC target value of $3×10^6$ and a maximum ion injection time of 20 ms. From the full MS scan, a maximum number of 35 of the most abundant precursor ions were selected for higher energy collisional dissociation (HCD) fragment analysis at a resolution of 15000 (at 200 m/z) with an automatic gain control (AGC) target value of $1×10^5$, a maximum ion injection time of 45 ms, a normalized collision energy of 32%, an intensity threshold of $8.3×10^3$, and the dynamic exclusion parameter set at 60 s. The resulting spectra from each run were searched separately against the SwissProt_2016_04 database with taxonomy of [human] database by the MaxQuant software (Computational Systems Biochemistry, Martinsried, Germany) package (version 1.5.1.2). The searched parameters are set as follows: mass tolerance for precursor ion was 10 ppm and mass tolerance for product ion was 0.02 Da. A maximum of 2 mis-cleavage sites were allowed. The identified protein contains at least 1 unique peptide. Carbamidomethyl was specified as fixed modifications. Oxidation of methionine (M) was specified as dynamic modification and acetylation. The identified Peptides and protein were retained and performed with FDR no more than 1.0%.

**RNA isolation and sequencing**. Firstly, RNA degradation and contamination was monitored on 1% agarose gels and RNA purity was checked using the Nano-Photometer® spectrophotometer (IMPLEN, CA, USA). Then, RNA integrity was assessed using the RNA Nano 6000 Assay Kit of the Bioanalyzer 2100 system (Agilent Technologies, CA, USA). A total amount of 1 µg RNA per sample from SNU5 cells were individually was used as input materials. Sequencing libraries were generated using NEBNext® UltraTM RNA Library Prep Kit for Illumina® (NEB, USA) following manufacturer's recommendations and index codes were added to attribute sequences to each sample. mRNA was purified in total RNA using poly-T oligo-attached magnetic beads. Fragmentation was carried out using divalent cations under elevated temperature in NEBNext First Strand Synthesis Reaction Buffer (5X). First strand cDNA was synthesized using random hexamer primer and M-MuLV Reverse Transcriptase (RNase H). Second strand cDNA synthesis was subsequently performed using DNA Polymerase I and RNase H. Remaining overhangs were converted into blunt ends via exonuclease/polymerase activities. After adenylation of 3' ends of DNA fragments, NEBNext Adaptor with hairpin loop structure were ligated to prepare for hybridization. In order to select cDNA fragments of preferentially 250–300 bp in length, the library fragments were purified with AMPure XP system (Beckman Coulter, Beverly, USA). After that, 3 µl USER Enzyme (NEB, USA) was used with size-selected, adaptor-ligated cDNA at 37 °C for 15 min following 5 min at 95 °C before PCR. PCR was performed with phusion high fidelity DNA polymerase, universal PCR primers and Index (X) primer. At last, PCR products were purified (AMPure XP system) and library quality was assessed on the Agilent Bioanalyzer 2100 system. Clustering of the index-coded samples was performed on a cBot Cluster Generation System using TruSeq PE Cluster Kit v3-cBot-HS (Illumia) according to the manufacturer's instructions. After cluster generation, the library preparations were sequenced on an Illumina Novaseq platform and 150 bp paired-end reads were generated.

Raw data (raw reads) were processed through in-house perl scripts. In this step, clean data (clean reads) were obtained by removing reads containing adapter, reads containing ploy-N and low-quality reads from raw data. At the same time, Q20, Q30 and GC content the clean data were calculated. All the downstream analyses were based on the clean data with high quality. Index of the reference genome was built using Hisat2 v2.1.0 and paired-end clean reads were aligned to the reference genome using Hisat2 v2.1.0. Counts v1.5.0-p3 was used to count the reads numbers mapped to each gene. Differential expression analysis of two groups was performed using the DESeq2 R package (1.16.1). The resulting P-values were adjusted using the Benjamini and Hochberg's approach for controlling the false discovery rate.

**Bioinformatics analysis**. Heat maps were produced using Perseus (1.6.2.2)[49]. Gene ontology and KEGG pathway analyses were performed using GeneCodis 3.0[50]. FDR (q value) was used to select interesting protein and gene sets.

**In vivo therapy assays of NR and SP group**. Four groups of female C57BL/6 mice ($n = 5$, 6–8-weeks-old) were subcutaneously implanted SNU5 cells for therapy. This study complied with relevant ethical regulations for animal testing and research, all animal protocols were approved by the Institutional Animal Care and Use Committee of the National Center for Nanoscience and Technology. The mice were randomly divided into four groups. When the tumors had been allowed to develop to approximately 100–200 mm³, mice were injected intravenously with AuNR-PEG-Ab (NR group), AuSP-PEG-Ab (SP group), Ab and PBS (100 µL) at a dose corresponding to 8 mg/kg of Ramucirumab ($n = 5$). PBS was served as control. Administration was carried out on once a week considering the drug was treated in GC patient once two weeks. The tumor sizes and weights were recorded

daily at the same time. Tumor sizes were measured by a vernier caliper. Tumor volume was calculated by the formula $(L \times W^2)/2$. L is for the longest and W is the shortest in tumor diameters (mm). The experimental data were assessed as the mean standard deviation using Origin software for four independent experiments. After that, tumors and other organs were collected. The toxicity in tumors and organs of each group were determined by H&E. The apoptosis of tumor cells was also detected by TUNEL assay, tissue slices were dewaxing and permeabilized, then TUNEL-FITC were incubated cells for 1 h at 37 °C and DAPI for 10 min. Ten images were acquired from three randomly selected sections from each tumor and evaluated for statistical analysis.

**Real-time polymerase chain reaction (RT-PCR) and western blot assay for checking the expression of FcγR.** Cells were cultured in the medium at the density of $10^7$ per dish. Total RNA was extracted using TRIzol reagent (Invitrogen) followed the manufacturer's instructions. RNA was reverse transcribed using the first strand synthesis kit (Takara) and subsequently cDNA was subjected to real-time PCR using SYBR green mix (Takara). PCR were performed on an ABI 7500fast (ABI Life Technologies). Ct Values obtained from the threshold cycle and were normalized by GAPDH amplified on the same cycle. The primers used were listed in Supplementary Table 2.

For western blot, cells were washed by PBS and lysed in lysis buffer (Sigma, USA) on ice for 30 min. Cell lysates were collected with centrifugation at 4°C for 30 min and the total protein was measured using BCA (bicinchoninic acid) protein determination assay kit (Beyotime Biotechnology, China). According to the standard western blot procedures, proteins were separated by SDS-PAGE and transferred onto PVDF (polyvinylidene fluoride) membranes (Merck Millipore, DE). The membranes were then blocked for 1 h in 5% non-fat milk in PBST and incubated with 1:1000 diluted anti-CD64 (EPR4623, abcam, UK) and 1:1000 diluted anti-CD16 (EPR22409-124, abcam, UK) antibodies and 1:2000 diluted β-actin (mAbcam 8226, abcam, UK) antibody respectively overnight at 4°C. After washing 3 times with PBST for 5 min each, the membranes were incubated with horseradish peroxidase-coupled isotype-specific secondary antibodies for 1 h at room temperature. The immune complexes were detected by ChemiDoc MP imaging system (Bio-Rad, USA).

**Cellular ROS measurement.** To assess cellular ROS levels, SNU5 cells were seeded at $5 \times 10^4$ cells per well in a 96-well plate for 24 h. then cells were treated by each component with the same concentration in MTT assay for 24. All treated cells were incubated with 20 μM DCFDA (Abcam) at 37 °C and 5% CO2 for 30 min. Fluorescence intensity was measured by Multi-scan Spectrum.

**Statistics and reproducibility.** Quantitative data are presented as mean ± SD. The differences between groups were compared using by unpaired, two-tailed independent student's t-test or one-way analysis of variance (ANOVA) with repeated measures followed by Tukey's HSD post hoc analysis was used for multiple comparison. No collected data were excluded from analysis. The p-value is noted either in the manuscript text or depicted in figures and legends as: $*P < 0.05$, $**P < 0.01$, $***P < 0.001$.

**Reporting summary.** Further information on research design is available in the Nature Research Reporting Summary linked to this article.

## Data availability

The data that support the findings of this study are available within the paper and its supplementary information files. The source data underlying Figs. 2c, d, 3g, h, 4a–h, i–l, 6a–e, 7a, b, d and Supplementary Figs. 1, 3 and 12 are provided as a Source Data file. The data that support he findings of this study are available from the corresponding author on request. Source data are provided with this paper. The proteomic raw data generated in this study have been deposited in the ProteomeXchange database under accession code P4. The transcriptomic raw data generated in this study have been deposited in the GEO database under accession code G9. Source data are provided with this paper.

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

## Acknowledgements

We acknowledge funding from the National Natural Science Foundation of China (22074006, 21775031, 81500900).

## Author contributions

M.Z. developed the concepts and designed the experiments; L.F., M.Z. Z.W. and W.W. performed the experiments; M.Z. and L.F. wrote the paper; M.Z. and L.F. analyzed results and edited the manuscript.

## Competing interests

The authors declare no competing interests.
