## [Peer Review File · Nature Communications]

Reviewers' Comments:

Reviewer #1 (Remarks to the Author):

In this manuscript the authors describe the results of experiments in which ramucirumab was altered by the addition of gold nanorods in an effort to enhance efficacy. Specifically, it was proposed that coupling ramucirumab to Peg functionalised nanorods and used in combination with Dox would increase tumour cell death in a gastric cancer model. Accordingly these changes would occur via the immune related phagocytosis by differential regulation of the Fc-gama receptor CD64 on the tumour cells.

While this is a worthy goal there are several aspects of the manuscript which require improvement. While there is potential for development here the current data do not support the claims made.

General comments:

- There are factual errors in the abstract and introduction – ramucirumab is not the first approved targeted therapy for gastric cancer and pembrolizumab is licensed for gastric cancer.
- The standard of English is not good. The paper is hard to follow and there are a lot of pieces of information missing, such as parts of protocols.
- There was no clear definition of the goals for the study.

Methodology:

There is a lot of information missing:

- Animal groups, number of animals per group, conditions studied
- Origin of reagents (such as Ramucirumab, PEG linker, DOX)
- Lack of description of the protocols presented
- Missing protocols (such as for the Ellman method, TEM ROS measurements, apoptosis, TUNNEL assay,...)

Gold nanoparticles

There is the need to present the chemistry of functionalisation (and methodology used in detail and stepwise).

Also, there was not described which amine has been coupled with both Cy5 or the PEG ligand using the EDC/NHS chemistry, and if this can affect the recognition ability of the Ab.

Cell culture

There is no agreement in concentration of VEGF concentrations used to treat HUVECs between the text in the methods and results sections.

In vivo experiments

There is no clear indication of mouse numbers, number of repeats, animals per group. There is also no clear description of the model: when are the collection days (after tumour injection), treatment schedules and tested concentrations, for each type of study. There are also contradictory sentences such as: "16 mice were randomly divided into four groups (n=5)".

Also, the supplementary methods are the same as the methods present in the main document.

Results:

The 220-850nm spectrum does not show the coupling of the antibody to the nanorods or nanoparticles. The AuSP-peg-Ab change in curve seems to be due to the existence of both nanoparticles and Ab and not the result of the correct coupling. There is the need to perform another characterisation technique for the correct composition of the constructions such as NMR or similar.

There is no quantification of the amount of Dox or Ab coupled to the AuSP or AuNR and how it was calculated the amount of nanoparticulate used for the in vitro and in vivo studies.

I liked the studies of biodistribution, although the authors do not address fully the impact of the toxicity of these nanoparticulates in the other tissues, mainly in the liver.

Figure 3 I-M do not identify the concentration used for the studies. For either DOX or Ab. And what does NC mean in the labelled graphs? PBS/no treatment, nano-carrier? Authors are presenting no effect of Ab treatment in this figure 3. Could this be due to the concentration used. Isn't it expected that these cells respond to treatment? Have the cells been checked for expression of the VEGFR2?

The author argue that NR are better at promoting apoptosis on tumour cells due to its geometry. Couldn't his be due to the different amounts of Ab present at the surface of each nano-carrier? It hasn't been shown any quantification for this. Couldn't this phenomenon also explain the results obtained in Figure 6A?

Figure 4 – Although I appreciate the pathway analysis of the altered genes and proteins it would have been beneficial to have the list of gene/proteins altered to understand the pursue of the phagosome pathway. In fact, the KEGG pathway presented in Figure 4M is completely illegible.

Figure 6D – the image is not clear at all. Also, they do not address liver toxicity (based on the mouse experiments). The authors should also have done tunnel assay in the liver tissue.

I do not believe the authors have not really showed that the different geometry is enough to improve therapy. Nevertheless, is interesting to see an immune-specific factor being expressed in tumour cells after antibody treatment.

Reviewer #2 (Remarks to the Author):

This manuscript described gold nanorod-based delivery system of Ramucirumab, and mechanism of enhanced effect of its anticancer effect by combining the gold nanorods. The combination of antibody and gold nanorods is unique and will be a promising strategy for enhancement of activity of antibody-based anticancer drug. The proteomics and transcriptomics analyses will also give us information of molecular mechanism of the enhancing effect of gold nanorods. However, fundamental molecular mechanism of the enhancement by rod-shaped nanoparticles, not by spherical nanoparticles, is still unclear. It is difficult to judge importance of this study that should be published in 'Nature Communications' at this stage.

Other points

1. In Figure 3, loading amount of DOX is also a key parameter for the cytotoxicity. Through the manuscript, amounts of nanoparticles, antibody, and DOX should be clearly indicated, and activities of different formulations should be carefully discussed by comparing these amounts.

2. In this study, DOX was conjugated with PEG via amide bond. It is stable linkage, and DOX is difficult to be released from PEG. Does such conjugated DOX show the cytotoxic activity?

3. In the line 124, is [Q2] typo of [Q3]? In Figure 3M, [Q3] is marked.

4. In the lines 130-131, the authors mentioned that [the higher toxicity of CTAB-coated spheres came from a higher release of toxic CTAB upon intracellular aggregation]. Generally, gold nanorods are prepared in aqueous solution containing CTAB to be the rod-shape, and CTAB still

remains on surface of the gold nanorods even after PEG modification. On the other hand, for gold nanosphere, CTAB is not necessary for their preparation. Or, the authors used gold nanospheres stabilized by CTAB? This part should be carefully discussed.

Detail Response to Reviewers

Thanks very much for reviewer's professional comments and advices. All the comments were responded carefully. The manuscript was revised per reviewer's advice and all the changes were highlight in red colour.

Reviewers' comments:

Reviewer #1:

In this manuscript the authors describe the results of experiments in which ramucirumab was altered by the addition of gold nanorods in an effort to enhance efficacy. Specifically, it was proposed that coupling ramucirumab to Peg functionalised nanorods and used in combination with Dox would increase tumour cell death in a gastric cancer model. Accordingly these changes would occur via the immune related phagocytosis by differential regulation of the Fc-gamma receptor CD64 on the tumour cells.

While this is a worthy goal there are several aspects of the manuscript which require improvement. While there is potential for development here the current data do not support the claims made.

General comments:

• There are factual errors in the abstract and introduction - ramucirumab is not the first approved targeted therapy for gastric cancer and pembrolizumab is licensed for gastric cancer.

Response:

Trastuzumab was approved by FDA for gastric cancer in 2010 [1]. The indications for breast cancer were metastatic breast cancer with high HER2 expression, and for gastric cancer were patients with metastatic gastric cancer with high HER2 expression. However, there are differences in medication methods. Breast cancer medication can be used alone; In gastric cancer, chemical drugs should be used together [2]. In 2014, FDA approved remolimumab (anti-vegfr2) as a single drug or in combination for the first line therapy treatment of metastatic or progressive gastric cancer [3, 4]. The major advantage of remolimumab is that it can make the disease stable again after the development of trastuzumab [5]. We apologized for the missing of precise attribute in the abstract. The refined description "the first approved single drug therapy for advanced gastric cancer" has been added in the revised manuscript. Besides, according to recent researches, about 36-40% GC patients expressed VEGFR2 while only 7-30% GC patients expressed HER2[6-10]. Thus, Trastuzumab also not be the best choice for gastric cancer without combines with chemical drugs although it comes earlier.

About pembrolizumab (Keytruda, anti-PD-1), currently the most widely used anti-PD-1 antibody drugs in the world are Keytruda (made by MSD in the United States, referred to as K drug) and Opdivo

(made by Bristol-Myers Squibb, referred to as O drug), which have effects on lung cancer, colorectal cancer, blood tumours and other diseases. However, in the clinical trial of gastric cancer, O drug all failed, while drug K succeeded in the study of KEYNOTE-059 [11], proving that drug K had a better objective remission rate than chemotherapeutic drug in the third-line treatment of gastric cancer. However, the failure in the KEYNOTE-061 and KEYNOTE-062 trials indicated that although K drug was safer than paclitaxel, it did not prolong overall survival (OS) or progression-free survival (PFS) in the first and second-line treatment. Thus, pembrolizumab may not be the first choice for gastric cancer.

All in all, although remolimumab is not the absolute first in time at every aspect, it is still the best choice for single use in gastric cancer until now. The related description had been refined in revised manuscript.

• The standard of English is not good. The paper is hard to follow and there are a lot of pieces of information missing, such as parts of protocols.

Response:

We are very sorry for the English expressing is not good enough for non-native speakers and apologized for the missing information which might be because of the negligence between different versions. In the revised manuscript, all missing information were completed and we had tried our best to refine the language.

• There was no clear definition of the goals for the study.

Response:

The original goal of this study was exactly not the shape specificity of gold nanorod. Our original goal was enhancing cytotoxicity and therapy efficiency of Ramucirumab. The involvement of gold nanorod and DOX beside Ab could act as the combining of target therapy, chemotherapy and phototherapy, which would synergistically abrogate tumours and prevent their recurrent, either with or without tumour resection, which had been described in detail in the revised manuscript. However, as the study proceeded, we found that the nano-carriers contained both Ab and gold nanoparticles can inhibit stomach tumour cell viability even without the exist of DOX, which was not showed in any reagent member alone except for DOX. Investigate the biological mechanism behind the enhanced tumour cell death for the nano-carriers became the advanced goal. The results suggested that these changes would occur via the immune related phagocytosis by differential regulation of the Fc-gamma receptor on the tumour cells, which was regarded as a worthy goal by reviewer. We believed that there were many researches which updated the goals when new findings appeared which were interesting and had not supposed in the origin.

Methodology:

There is a lot of information missing:

- Animal groups, number of animals per group, conditions studied

Response:

We apologized for the missing and omitted information. We thought some of them were showed in figures so it was not showed detailed enough in methodology section. In the revised manuscript, all missing information were completed.

- Origin of reagents (such as Ramucirumab, PEG linker, DOX)

Response:

We are sorry for the missing, and this information include reviewer mentioned below have been added in the revised manuscript.

Ramucirumab (Cyramza, Lilly), HS-PEG3500-COOH (Nanosoft), Doxorubicin (Zhejiang Hisun Pharmaceutical Co)

- Lack of description of the protocols presented

- Missing protocols (such as for the Ellman method, TEM ROS measurements, apoptosis, TUNNEL assay, ...)

Response:

For some common assays, we usually use compact description or cite references to the common treatment from some major journals. We apologized for the missing to some protocols which might come from the neglect of method of late added operations or the missing between different versions of manuscript. The protocols have been completed and improved in the revised version.

-Gold nanoparticles

There is the need to present the chemistry of functionalisation (and methodology used in detail and stepwise).

Also, there was not described which amine has been coupled with both Cy5 or the PEG ligand using the EDC/NHS chemistry, and if this can affect the recognition ability of the Ab.

Response:

The functionalization method referenced from a published work from Nature Materials [12] and the description had been refined to detailed per it in the revised manuscript. We will add characterization if necessary when the public health crisis about new coronavirus 2019-nCoV in China relieved. The Ab was coupled to PEG-gold nanorods by a carbodiimide chemistry assisted by N-hydroxy succinimide (EDC/NHS coupling reaction) between the carboxylated PEG terminal or Cy5 and the primary amine groups of the antibody. The same coupling method was used to each group, so the influence to recognition ability of the Ab would perform equal to these groups. And the results in vitro and in vivo confirmed the recognition ability of the Ab after functionalization.

-Cell culture

There is no agreement in concentration of VEGF concentrations used to treat HUVECs between the text in the methods and results sections.

Response:

We felt very sorry for the mistake and the right concentration 20 ng/mL had been updated in the revised version. At this concentration, the cell viability of HUVECs is highest to 24 h treatment (Figure S6).

-In vivo experiments

There is no clear indication of mouse numbers, number of repeats, animals per group. There is also no clear description of the model: when are the collection days (after tumour injection), treatment schedules and tested concentrations, for each type of study. There are also contradictory sentences such as: “16 mice were randomly divided into four groups (n=5)”.

Response:

We apologized for the missing, mistake and oversimplification to some information in Figure 6. The number of repeats was mentioned in the statistics part of method section. The corrected and detailed description about in vivo experiments were supplied in revised version.

-Also, the supplementary methods are the same as the methods present in the main document.

Response:

This was prepared in case of the main content exceeded word limit and we thought the supplementary material didn't occupied the length of main body. If reviewer thought it was reduplicative, we will delete the same part with main body and add the detailed and stepwise description about the functionalization process of nano-carriers in the supplementary methods.

-Results:

The 220–850nm spectrum does not show the coupling of the antibody to the nanorods or nanoparticles. The AuSP-peg-Ab change in curve seems to be due to the existence of both nanoparticles and Ab and not the result of the correct coupling. There is the need to perform another characterisation technique for the correct composition of the constructions such as NMR or similar.

Response:

In the results of TEM it could be seen that a clear layer on the surface of gold nanomaterials, similar with the reference [12]. Together with absorbance spectrum and the cytotoxicity test in Figure 3 indicated that neither gold nanomaterials nor Ab don't show toxicity in tumour cell at a very wide concentration range. If these parts hadn't reacted and coupled, the cytotoxicity would be the same with their alone performance. We will add another characterization per reviewer's advice if the public health crisis about new coronavirus 2019-nCoV in China relieved.

-There is no quantification of the amount of Dox or Ab coupled to the AuSP or AuNR and how it was calculated the amount of nanoparticulate used for the in vitro and in vivo studies.

Response:

For different groups, it can't be achieved to make all reagents have the same concentration. And it was hardly to quantification the amount of Dox or Ab coupled to the AuSP or AuNR. Thus, the same amount of Ramucirumab antibody (Ab) was used, for Ab is the core for cancer therapy. And the left reagents included the amount of nanoparticulate within different groups were calculated per the same amount of Ab. That is, add each reagent per certain ratio, and when accomplished all the functionalization, detected the concentration of Ab (protein concentration method) in each nano-carrier and decide the final amount of each group via the same amount of Ab. The detailed method and values had been added in the revised method section and supplementary methods.

-I liked the studies of biodistribution, although the authors do not address fully the impact of the toxicity of these nanoparticles in the other tissues, mainly in the liver.

Response:

Thanks for reviewer's comment. In the results of biodistribution, the highest fluorescence intensity appeared in liver except for tumour region, while other tissues were very low. And liver was regarded as the main detoxification tissue. Ab has been approved by FDA and gold nanoparticle was recognized as a biocompatible nanomaterial. So other tissues were not assessed globally. The spread of the new coronavirus 2019-nCoV in China make it hardly to do experiment because most of people were restricted activity at presently.

-Figure 3 I-M do not identify the concentration used for the studies. For either DOX or Ab. And what does NC mean in the labelled graphs? PBS/no treatment, nano-carrier?

Response:

We are sorry for the missing and the concentration for these reagents have been added in the revised manuscript. NC means PBS treated group.

-Authors are presenting no effect of Ab treatment in this figure 3. Could this be due to the concentration used. Isn't it expected that these cells respond to treatment? Have the cells been checked for expression of the VEGFR2?

Response:

In Figure 3B-D, a wide range of Ab were used in three kind of cells, so the no effect should not be due to low concentration. It was not expected that these cells respond to Ab treatment except HUVEC. So, when the coupling complex induced inhibition response in gastric cancer cell, it became interesting finding of this study. The cells had been checked for expression of the VEGFR2 by flow cytometry (**Figure 5F** and **Figure S5. A-B**). The responses may also relate with the VEGFR2 expression in gastric cancer cell. No expression of VEGFR2 in normal stomach cell GES-1 and another kind of cancer cell

Hela may also indicate that this direct cytotoxicity does not exist in normal stomach cell which was confirmed in **Figure 5B-C** and Hela.

-The author argue that NR are better at promoting apoptosis on tumour cells due to its geometry. Couldn't his be due to the different amounts of Ab present at the surface of each nano-carrier? It hasn't been shown any quantification for this. Couldn't this phenomenon also explain the results obtained in Figure 6A?

Response:

The different amounts of Ab don't influence gastric tumour cell apoptosis in the free form (Figure 3B-C). The most important was the existing of both antibody and gold nanoparticles. Both gold nanorod and gold nanosphere induced such cell killing effect and immune related phagocytosis response than blank group from our indirect comparison of them (AuNR-Peg-Ab and AuSP-Peg-Ab) to blank cell in transcriptomics result. But gold nanorod induced more and stronger response in the proteomic and transcriptomics results for direct comparison, as well as stronger regulation of Fc-gamma receptor. So, this apoptosis promoting phenomenon is not only correlated with geometry, but the priority of gold nanorod still due to its geometry. Reviewer's opinion is reasonable, However, supposed that this phenomenon is due to the different amounts of Ab present at the surface of each nano-carrier, whether the different carrying amounts of Ab also be caused by different geometry? And isn't the geometry still the original reason for its priority? Actually, in this study, amounts of Ab or DOX was measured when each complex material had been synthesis. Thus, amounts of Ab in each nano-carrier treated cells were the same. Actually, our results had proved that the intracellular concentration of Ab in GC cell was nearly the same for both AuNR-peg-Ab (NR group) and AuSP-peg-Ab (SP group) by flow cytometry (**Figure 2E**). We also consider that this apoptosis promoting phenomenon explains the results obtained in **Figure 6A**.

-Figure 4 - Although I appreciate the pathway analysis of the altered genes and proteins it would have been beneficial to have the list of gene/proteins altered to understand the pursue of the phagosome pathway. In fact, the KEGG pathway presented in Figure 4M is completely illegible.

Response:

Thanks for reviewer's advice. However, it could be seen in Figure 4A that there had such many proteins and genes which only expressed in one group, and these were regarded as altered. And for the middle part which showed co-expression, there also hundreds of genes and proteins which showed up- or down-regulation. Thus, it was too large for main body. Currently, most papers about system biology analysis mainly display figure format. And the whole list of these genes and proteins had been supplied in the supplementary materials along with their expression values. If reviewer want to know the reason

for the pursue of the phagosome pathway, it is because the large amount of altered gene and proteins enriched in this pathway and especially the high p value of enrichment analysis (**Figure 4K-L**). for pursuing the detail of the phagosome pathway, Figure 4M indeed displayed the specific members in this pathway. We had tried our best to make it clear and when you enlarge the word document to 400 folds you can see it clearly. However, it needs to be contained in Figure 4 and could not to be seen clearly in the whole figure. In the revised version, the original file of Figure 4M has been attached.

-Figure 6D - the image is not clear at all. Also, they do not address liver toxicity (based on the mouse experiments). The authors should also have done tunnel assay in the liver tissue.

Response:

The original and high-resolution picture of Figure 6D would be supplied alone. Because It takes up a small part of the Figure 6, it was hard to make it too clear. Usually tunnel assay was not do in many tissue and haematoxylin-eosin(H&E) staining is also a classical assay for tissue damage. We will consider to supplement this assay if necessary per reviewer's advice after the popular of the new coronavirus 2019-nCoV in China.

-I do not believe the authors have not really showed that the different geometry is enough to improve therapy. Nevertheless, is interesting to see an immune-specific factor being expressed in tumour cells after antibody treatment.

Response:

I am sorry for not very understand the first sentence because two not were used. However, in the proteomic and transcriptomic analyses, from direct comparison of rod to spherical nanoparticles, and indirect comparisons of them compared with blanks cells with no treatment, these changes between the direct and indirect comparisons were all enriched in immune related phagocytosis. And these two shaped nano-gold induced differential regulation of the Fc-gamma receptor on the tumour cells. Both gold nanorod and nanosphere induced such response, but gold nanorod had more and stronger regulation, so therapy improvement is not only correlated with geometry. And the most important was the existing of both antibody and gold nanoparticles. But rod-shaped induced more response and regulation than sphere-shaped group. Together with the same loading concentration of Ab, it could be deduced that geometry was the original element to more tumour killing effect of gold nanorod than nanosphere.

Reviewer #2 (Remarks to the Author):

This manuscript described gold nanorod-based delivery system of Ramucirumab, and mechanism of enhanced effect of its anticancer effect by combining the gold nanorods. The combination of antibody and gold nanorods is unique and will be a promising strategy for enhancement of activity of antibody-based anticancer drug. The proteomics and transcriptomics analyses will also give us information of molecular mechanism of the enhancing effect of gold nanorods. However, fundamental molecular mechanism of the enhancement by rod-shaped nanoparticles, not by spherical nanoparticles, is still unclear. It is difficult to judge importance of this study that should be published in 'Nature Communications' at this stage.

Response:

From direct comparison of rod and spherical nanoparticles included groups in the proteomic and transcriptomic analyses, and indirect comparisons of them compared with blank cells with no treatment, the results showed that these changes would occur via the immune related phagocytosis. And the following assay confirmed the difference in tumour cell inhibition effect mainly came from differential regulation of the Fc-gamma receptor on the tumour cells.

Both gold nanorod and gold nanosphere induced such cell killing effect and immune related phagocytosis response than blank group from our indirect comparison of them (AuNR-Peg-Ab and AuSP-Peg-Ab) to blank cell in transcriptomics result. But gold nanorod induced more and stronger response in the proteomic and transcriptomics results for direct comparison, as well as stronger regulation of Fc-gamma receptor. The most important was the existing of both antibody and gold nanoparticles, but rod-shaped group induced more regulation than sphere-shaped group. These two groups with different shape of gold nanoparticles induced the same kind of response of stomach tumour cell with different levels. Together with the same final concentration of Ab in each treatment group used in this study, geometry should be an original element to more tumour killing effect of gold nanorod than nanosphere.

Other points

1. In Figure 3, loading amount of DOX is also a key parameter for the cytotoxicity. Through the manuscript, amounts of nanoparticles, antibody, and DOX should be clearly indicated, and activities of different formulations should be carefully discussed by comparing these amounts.

Response:

We apologized for the missing of amounts in the tests after the IC50 curve. The amount of each reagent was calculated per constant ratio and the same amount of antibody (8 µg/ml according to human recommended dose) after preparation. And the missing amounts had been supplied in the revised version. However, although we agreed that loading amount of DOX is important, the absolute amount of DOX doesn't influence the key finding in this manuscript. As in the results of cytotoxicity of single agent showed nearly no toxicity up to very high concentration except for DOX, which showed obvious cytotoxicity in a relative low concentration (Figure 3A-H, I, K). In addition, in the nano-delivery

system assessment which contain complex of nanoparticles, antibody, and DOX (Figure 3J, L), the comparison of cytotoxicity were between every two groups with the similar composition like AuNR-Ab vs. AuSP-Ab, AuNR-DOX vs. AuSP-DOX and AuNR-Ab-DOX vs. AuSP-Ab-DOX. The DOX amount of each comparison were the same. And among different pairs, the amount of DOX were the same per the same loading amount of Ab if DOX was include in certain groups. Actually, this assay mainly intended to find the role of nanoparticles in the direct cytotoxicity to gastric tumour cell when they were combined with Ab. And the difference of cytotoxicity came from whether DOX existed (among different pairs) or not and the difference of shape of gold nanoparticles (within comparison pairs), but not the amount of DOX. The use of DOX could act as the combining of chemotherapy. And the conclusion could also be addressed without the existing DOX. Antibody (Ab) was chosen as reference because it is the core of target therapy. Then with the same concentration of Ab, nanoparticles and DOX within comparison pair (the loading ratio of different reagent is fixed), the difference of cytotoxicity within comparison pair mainly attributed to the shape of nanoparticles.

2. *In this study, DOX was conjugated with PEG via amide bond. It is stable linkage, and DOX is difficult to be released from PEG. Does such conjugated DOX show the cytotoxic activity?*

Response:

From the results Figure 3J and Figure 3L, it can be seen that the groups included DOX showed more cytotoxic activity to relevant groups without DOX.

3. *In the line 124, is [Q2] typo of [Q3]? In Figure 3M, [Q3] is marked.*

Response:

Yes, the correction had been made in the revised version.

4. *In the lines 130–131, the authors mentioned that [the higher toxicity of CTAB-coated spheres came from a higher release of toxic CTAB upon intracellular aggregation]. Generally, gold nanorods are prepared in aqueous solution containing CTAB to be the rod-shape, and CTAB still remains on surface of the gold nanorods even after PEG modification. On the other hand, for gold nanosphere, CTAB is not necessary for their preparation. Or, the authors used gold nanospheres stabilized by CTAB? This part should be carefully discussed.*

Response:

Both gold nanorods and gold nanosphere used in this study were commercialized. Both of them were not CTAB-capped and the impurities of them were very low and there weren't CTAB existed per instruction books from manufacturers. And the CTAB mentioned was just for the certain situation in that reference. If it created confusion we can delete it in the revised version. What the most important is, both gold nanorods and gold nanosphere showed no cell toxicity respectively when they were

loading alone in the study. Thus, their toxicity after linked with PEG and antibody would not attribute to the coat of gold nanorods or gold nanosphere themselves.

References

- 1 Alice Y.S. Law & Wong, C. K. C. Stanniocalcin-1 and -2 promote angiogenic sprouting in HUVECs via VEGF/VEGFR2 and angiopoietin signaling pathways. *Molecular & Cellular Endocrinology* **374**, 73-81 (2013).
- 2 Rebischung, C., Barnoud, R., Stefani, L., Faucheron, J. L. & Mousseau, M. The effectiveness of trastuzumab (Herceptin) combined with chemotherapy for gastric carcinoma with overexpression of the c-erbB-2 protein. *Gastric Cancer* **8**, 249-252, doi:10.1007/s10120-005-0342-7 (2005).
- 3 Casak, S. J. *et al.* FDA Approval Summary: Ramucirumab for Gastric Cancer. *Clinical Cancer Research* **21**, 3372-3376 (2015).
- 4 Casak, S. J. *et al.* FDA Approval Summary: Ramucirumab for Gastric Cancer. *Clin Cancer Res* **21**, 3372-3376, doi:10.1158/1078-0432.CCR-15-0600 (2015).
- 5 Tehfe, M., Tabchi, S., Laterza, M. M. & Vita, F. Ramucirumab in HER-2-positive gastroesophageal adenocarcinoma: an argument for overcoming trastuzumab resistance. *Future Oncology* **14**, 223 (2018).
- 6 Chan, S. L. & So, J. B. Gastric Cancer Therapy. 1845-1852 (2016).
- 7 Rebischung, C., Barnoud, R., Stefani, L., Faucheron, J. L. & Mousseau, M. The effectiveness of trastuzumab (Herceptin) combined with chemotherapy for gastric carcinoma with overexpression of the c-erbB-2 protein. *Gastric Cancer* **8**, 249-252, doi:10.1007/s10120-005-0342-7 (2005).
- 8 Willert, E. K., Rajagopalan, S., Robinson, G. L., Brieschke, B. & Higgins, J. P. J. C. R. Abstract P4-15-17: A novel targeted engineered toxin body for treatment of HER2 positive breast cancer. *Cancer Research* **75**, P4-15-17-P14-15-17 (2015).
- 9 Bonelli, P. *et al.* Precision medicine in gastric cancer. *World J Gastrointest Oncol* **11**, 804-829, doi:10.4251/wjgo.v11.i10.804 (2019).
- 10 Aprile, G. *et al.* The challenge of targeted therapies for gastric cancer patients: the beginning of a long journey. *Expert Opin Investig Drugs* **23**, 925-942, doi:10.1517/13543784.2014.912631 (2014).
- 11 Fuchs, C. S. *et al.* Safety and Efficacy of Pembrolizumab Monotherapy in Patients With Previously Treated Advanced Gastric and Gastroesophageal Junction Cancer: Phase 2 Clinical KEYNOTE-059 Trial. *JAMA Oncol* **4**, e180013, doi:10.1001/jamaoncol.2018.0013 (2018).
- 12 Conde, J. O., Oliva, N., Zhang, Y. & Artzi, N. J. N. M. Local triple-combination therapy results in tumour regression and prevents recurrence in a colon cancer model.

REVIEWER COMMENTS

Reviewer #2 (Remarks to the Author):

The manuscript was revised according to the reviewers' comments. However, some points haven't been clarified, yet.

1. In the response for the comment 2 from the reviewer #2, the authors responded that 'In this study, DOX was not conjugated with PEG but mixed directly with the same ratio/concentration in different nano-carriers'. However, the authors mentioned that 'Ramucirumab and/or doxorubicin functionalized with Au-Peg by standard EDC/NHS coupling reaction' in the line 338-339. It means that DOX was covalently coupled on Au-PEG. This part should be carefully indicated. In addition, If DOX bound on Au-PEG non-covalently, what is its interaction? How the binding stability? It is an important factor for its delivery.

2. Relating to the previous comment, when I confirmed the method for the Cy5-labeling of Ramucirumab and/or doxorubicin, again, I noticed that it is still difficult to understand what the authors did. English not only in this part but also through the main text should be revised. At least, abbreviation of 'PEG', 'Peg', and 'peg', should be unified.

Reviewer #3 (Remarks to the Author):

In this manuscript the authors describe the results of experiments in which ramucirumab was altered by the addition of gold nanorods in an effort to enhance efficacy. I have appreciated the work of the authors to revise the paper, but I found the paper not suitable, for example in the introduction section there is an error as ramucirumab has been approved in second line of treatment.

Detail Response to Reviewers

Thanks very much for reviewer's professional comments and advice. All the comments were responded carefully. The manuscript was revised per reviewer's advice and all the changes were highlight in red colour.

Reviewers' comments:

Reviewer #2 (Remarks to the Author):

The manuscript was revised according to the reviewers' comments. However, some points haven't been clarified, yet.

1. In the response for the comment 2 from the reviewer #2, the authors responded that 'In this study, DOX was not conjugated with Peg but mixed directly with the same ratio/concentration in different nano-carriers'. However, the authors mentioned that 'Ramucirumab and/or doxorubicin functionalized with Au-Peg by standard EDC/NHS coupling reaction' in the line 338-339. It means that DOX was covalently coupled on Au-Peg. This part should be carefully indicated. In addition, If DOX bound on Au-Peg non-covalently, what is its interaction? How the biding stability? It is an important factor for its delivery.

Response:

We felt very sorry for missing this place of DOX description when we revised manuscript after we make this question clear. We have corrected this description in the latest revised version and in the latest version, it also showed 'DOX was mixed directly with the same concentration (0.3µg/mL) for the groups possessing DOX.' in the line 340-341. In cancer nanomedicine, DOX conjugation or mixing with polymers has been used to reduce its side effects and improve the solubility and bioavailability of this hydrophobic drug. In this study, DOX was mixed and adsorbed on the AuNP-Peg by static electricity interaction at a saturation condition. Our original goal was enhancing therapy efficiency of target therapy. The involvement of gold nanorod and DOX beside Ramucirumab could act as the combining of target therapy, chemotherapy and photo-therapy. We don't aim to increase the delivery efficiency of DOX within chemotherapy. The existing of Ab may also influence the interaction of DOX and Peg. However, the most important is that along with the study proceeded, we found that the nano-carriers contained both Ab and gold nanoparticles can inhibit stomach tumour cell viability even

without the existing of DOX. Investigation the biological mechanism behind the enhanced tumour cell death for the nano-carriers became the advanced goal. DOX was only used in the in vitro tests and all in vivo studies (imaging, tumour therapy and multi-omics studies) didn't involve DOX.

2. Relating to the previous comment, when I confirmed the method for the Cy5-labeling of Ramucirumab and/or doxorubicin, again, I noticed that it is still difficult to understand what the authors did. English not only in this part but also through the main text should be revised. At least, abbreviation of 'PEG', 'Peg', and 'peg', should be unified.

Response:

We apologize for the confusion made by the language and we tried to refine them and abbreviation carefully in the latest revised version. In order to help reviewer to understand the material synthesis process, we add a schematic diagram here (also add in supporting information Figure S13).

Figure R1 Schematic diagram of material synthesis process (All drawings are not in scale)

Reviewer #3 (Remarks to the Author):

In this manuscript the authors describe the results of experiments in which ramucirumab was altered by the addition of gold nanorods in an effort to enhance efficacy. I have appreciated the work of the authors to revise the paper, but I found the paper not suitable, for example in the introduction section there is an error as ramucirumab has been approved in second line of treatment.

Response:

We are very sorry for the mistake and have deleted the incorrect statement and refined relate description in the latest revised version. Although ramucirumab has been approved only in second line

of treatment, it is the only one which approved initially monotherapy to for previously treated patients with advanced or metastatic gastric or gastroesophageal junction adenocarcinoma.

REVIEWERS' COMMENTS

Reviewer #2 (Remarks to the Author):

The reviewers' comments were reflected in the revised manuscript. It is acceptable for publicatio

Detail Response to Reviewers

Thanks very much for reviewer's professional comments and advice. All the comments were responded carefully.

Reviewers' comments:

Reviewer #2 (Remarks to the Author):

The reviewers' comments were reflected in the revised manuscript. It is acceptable for publication.

Response:

We thank the reviewer for the kind comments!